# Impact of symmetry in local learning rules on predictive neural representations and generalization in spatial navigation

Janis Keck[1,2,3]*, Caswell Barry[4‡], Christian F. Doeller[2,3,5‡], Jürgen Jost[1,3,6,7‡]

**1** Max Planck Institute for Mathematics in the Sciences, Leipzig, Germany, **2** Max Planck Institute for Human Cognitive and Brain Sciences, Leipzig, Germany, **3** Max Planck School of Cognition, **4** Department of Cell and Developmental Biology, University College London, London, WC1E 6BT, UK, **5** Kavli Institute for Systems Neuroscience and Jebsen Centre for Alzheimer's Disease, Norwegian University of Science and Technology, Trondheim, Norway , **6** ScaDS.AI - Center for Scalable Data Analytics and Artificial Intelligence, Leipzig, Germany, **7** Santa Fe Institute for the Sciences of Complexity, Santa Fe, New Mexico, USA

‡ Shared senior authorship.
* janis.keck@maxplanckschools.de

**Data availability statement:** Code used for experiments and to generate the data used therein is publicly available at https://github.com/jakeck1/sympredlearning.

## Abstract

In spatial cognition, the Successor Representation (SR) from reinforcement learning provides a compelling candidate of how predictive representations are used to encode space. In particular, hippocampal place cells are hypothesized to encode the SR. Here, we investigate how varying the temporal symmetry in learning rules influences those representations. To this end, we use a simple local learning rule which can be made insensitive to the temporal order. We analytically find that a symmetric learning rule results in a successor representation under a symmetrized version of the experienced transition structure. We then apply this rule to a two-layer neural network model loosely resembling hippocampal subfields CA3 - with a symmetric learning rule and recurrent weights - and CA1 - with an asymmetric learning rule and no recurrent weights. Here, when exposed repeatedly to a linear track, neurons in our model in CA3 show less shift of the centre of mass than those in CA1, in line with existing empirical findings. Investigating the functional benefits of such symmetry, we employ a simple reinforcement learning agent which may learn symmetric or classical successor representations. Here, we find that using a symmetric learning rule yields representations which afford better generalization, when the agent is probed to navigate to a new target without relearning the SR. This effect is reversed when the state space is not symmetric anymore. Thus, our results hint at a potential benefit of the inductive bias afforded by symmetric learning rules in areas employed in spatial navigation, where there naturally is a symmetry in the state space.

## Author summary

The hippocampus is a brain region which plays a crucial role in spatial navigation for both animals and humans. Contemporarily, it's thought to store predictive representations of the environment, functioning like maps that indicate the likelihood of visiting

**Funding:** JK, JJ and CFD are supported by Max-Planck Institute for Human and Cognitive Brain Sciences. JK and JJ are supported by Max-Planck Institute for Mathematics in the Sciences. JK, JJ and CFD are supported by Max-Planck School of Cognition. JJ is supported by ScaDS.AI Leipzig. CB is funded by a Wellcome SRF. The funders had no role in study design, data collection and analysis, decision to publish, or preparation of the manuscript.

**Competing interests:** The authors have declared that no competing interests exist.

certain locations in the future. In our study, we used an artificial neural network model to learn these predictive representations by adjusting synaptic connections between neurons according to local learning rules. Unlike previous research, our model includes learning rules that are invariant to the temporal order of events, meaning they are symmetric with respect to the reversal of input timings. This approach produces predictive representations particularly useful for understanding spatial relationships, as navigating from one point to another is often equivalent to the reverse. Our model offers additional insights: it replicates observed properties of hippocampal cells and helps an artificial agent solve navigation tasks. The agent trained with our model not only learns to navigate but also generalizes better to new targets compared to traditional models. Our findings suggest that symmetric learning rules enhance the brain's ability to create useful predictive maps for problems which are inherently symmetric, as is navigation.

## 1. Introduction

The hippocampus and its adjacent sub- and neocortical regions are widely believed to form both a crucial part in the acquisition and storage of memory, as well as the encoding of spatial and navigational variables in the form of spatially stable neural responses [1–5].

It is an increasingly popular assumption that the representations which the brain generates in general, and in particular for space and memory, are not merely descriptive of the current state of the world, post-dictions of events or places just passed. Rather, it is believed that a predictive representation is learned, such that the objective is to infer future states of the world from one's experience [6–10].

One framework that has extensively been used to describe this objective on the algorithmic level comes from reinforcement learning. The so called 'successor representation' (SR), or more broadly 'successor features' (SF) are a generalization of the well known value function, and are essentially a conditional expectation: Given the current state, they encode a (weighted) expectation of future values of a given function of the states of the world [11,12]. If that function is simply an indicator of the states, then one obtains the SR, which thus roughly encodes how often states will be visited in the future.

Originally, the successor representation was proposed as an intermediate between 'model-based' and 'model-free' reinforcement learning [13,14], allowing the storage of certain information about the transition structure under a given policy - hence, affording some generalization to different reward structures - while still being possible to learn with a efficient temporal difference (TD) learning algorithm [9,11]. Later work has also used the SR for different objectives such as option discovery [15] and reward free exploration [16]. More generally, maintaining a predictive representation might be a useful feature of intelligent agents (biological or artificial ones) that have to plan their behaviour [17].

In the hippocampal navigation literature, the successor representation view has been influential because apart from fitting well with the more general predictive brain hypothesis, it could explain non-trivial effects of place cells that had been previously observed, for example the skewing of place fields in direction of travel or the non-extension of place-fields through obstacles in the environment [8]. Furthermore, successor representation theory yielded an algorithmic explanation for grid cells as an eigendecomposition of place-cell structure, which could also be connected to efficient neurally plausible navigation [18–21].

Despite the success of SR theory explaining neural data on an algorithmic level, there has been considerably less work dedicated to providing a mechanism through which the SR should be learned using biologically plausible learning rules [22]. Recently, this question

has been tackled by the community: Two papers [23,24] used feedforward networks to learn synaptic weights that compute successor features from their inputs. [23] focuses on the predictiveness afforded by the theta cycle together with a compartment-neuron learning rule, while [24] uses spiking neural networks and a spike-time-dependent plasticity (STDP) rule. On the other hand, [25] used a recurrent neural network, to learn successor features directly in the activities of the recurrently connected neurons.

Anatomically, the latter approach can be linked to plasticity occurring at the recurrent synapses of CA3, while the former approach maps on the feedforward synapses to CA1 [26]. Both areas are known to show a considerable proportion of place cells [27], hence both are indeed candidate regions to encode successor representations. However, it has been suggested that different learning rules might be in place at the respective synapses: the Schaffer collateral- synapse to CA1 pyramidal cells is classically believed to obey the rules of STDP [28,29], which in its stereotypical form requires presynaptic increased activity to precede postsynaptic increased activity for an increase in strength of synaptic connection [30,31]. On the other hand, recent work has identified a regime in which recurrent CA3 synapses get strengthened if pre- and postsynaptic increased activity are close in time, regardless of the temporal order - and computationally linked a symmetric learning rule to benefit in memory storage of a recurrent network [32].

Here, we want to investigate the effect of such symmetric learning rules on the construction of predictive representations. That is, we aim to understand whether using a learning rule insensitive to the temporal order of the inputs learns different successor representations - and which (dis-)advantages it yields.

To this end, we first construct a model which has both a recurrent and a feedforward component, reminiscent of the architecture of Hippocampus, and study the successor features that are learned using a local learning rule. Thereby we extend the earlier work which focused on learning in a single layer to learning at multiple levels. This extension is rather straightforward and results in both layers learning successor features based on their respective inputs.

We then find that by changing the learning rule, the representations also undergo a similar modification: In the symmetric setting, instead of encoding future expectations under the current true policy of the agent, successor features under a symmetrized version of the transition probabilities are learned, while an asymmetric rule learns the 'true' successor features. We then contrast the utility of the respective representations in a reinforcement learning setting. There, we find that a symmetric learning rule yields benefits for generalization in navigational tasks, where the symmetry of the state space can be exploited, while an asymmetric learning rule is more advantageous for generalization in asymmetric state spaces. We conclude that implementing both an asymmetric and a symmetric learning rule might yield complementary representations.

## 2. Results

### 2.1. Successor representations

The successor representation and the more general successor features describe future expectations of a quantity, conditional on the current state of the world. They are most easily defined in the following setting: Assume the environment of an agent/animal consists of a set of states $\mathcal{S}$. The states of the world are changing according to a time homogeneous Markov chain, denoted $S_t \in \mathcal{S}$, with transition probabilities encoded in the matrix $P$ such that $P_{s,s'} := p(s'|s)$, the probability transitioning from state $s$ to state $s'$ in one timestep. Then for any

feature/observation function

$$\phi : \mathcal{S} \to \mathbb{R}^m \qquad (1)$$

one can define an expectation of weighted, cumulative future values of that function, given the current state:

$$SF_\phi(s) = \mathbb{E}\left[\sum_{k=0}^{\infty} \gamma^k \phi(S_{t+k}) \,\Big|\, S_t = s\right]. \qquad (2)$$

The weighting factor $\gamma \in [0, 1)$ puts relatively more importance on proximal times. In the case that $\phi$ is an injective function, that is for every state of the world there is an unique observation value, it makes sense to define the 'successor representation' (SR)

$$SR_\phi(\nu) = \mathbb{E}\left[\sum_{k=0}^{\infty} \gamma^k \phi(S_{t+k}) \,\Big|\, \phi(S_t) = \nu\right]. \qquad (3)$$

We use this terminology here in a little more generality than is usual, but it can be seen that one important special case leads to what is usually called SR: Let $\phi^e(s) := e_s$, where $e_s$ is the unit vector with a 1 at the entry corresponding to state $s$. That is, $\phi^e$ assigns to each state its indicator vector. Then one obtains

$$SR_{\phi^e}(e_s) = \sum_{k=0}^{\infty} \gamma^k \sum_{s' \in \mathcal{S}} P_{s,s'}^k e_{s'} \qquad (4)$$

$$= \sum_{k=0}^{\infty} \gamma^k e_s^T P^k = (Id - \gamma P)_s^{-1}, \qquad (5)$$

where the last equality is the well known identity for the Neumann series. The matrix $(Id - \gamma P)^{-1}$ is widely referred to as the SR, so our definition encompasses this special case. One can see from these equations that the SR of indicator vectors thus gives a weighted sum of expected future visitation probabilities of states, and it is this expression that has originally been used to model the predictive representations that hippocampus ought to encode [8,9].

## 2.2. Learning successor representations with local learning rules

We construct a simple model of two neuron population activities $p_t$ which have dynamics of the form

$$\frac{d}{dt}p^1 = -p^1 + \sigma(\gamma_1 W^r p^1 + (1 - \gamma_1)\phi^1) \qquad (6)$$

$$\frac{d}{dt}p^2 = -p^2 + \sigma(\gamma_2 W^f p^1 + (1 - \gamma_2)\phi^2). \qquad (7)$$

Here, $W^r$ is a recurrent connectivity matrix that feeds the activity of the first population back into itself, while $W^f$ is feedforward matrix, which encodes how activity of the first population is fed into the second. This architecture of one population of highly recurrently connected neurons feeding into a second one with little recurrent connectivity is reminiscent of hippocampal subfields CA3 and CA1 respectively [26]. The two populations obtain additional external inputs $\phi^1, \phi^2$, which might represent the input to CA3 via mossy fibers, or directly through the perforant path, and the input to CA1 from EC through the latter, respectively. In the simplest case, these inputs are just indicator-functions for particular states, that is $\phi_i(s) = \delta(s = s_i)$. A more realistic shape, which we employ in our experiments in Sect 2.5, might be

given by Gaussian inputs of the form $\phi_i(s) = \exp\left(-\frac{\|s-\mu_i\|^2}{2\sigma}\right)$. The $\gamma_i \in [0, 1]$ are scalar-valued global gain factors which control the relative strengths of inputs to the populations. Note that we intentionally used the same symbol $\gamma$ as for the timescale factors in the successor representation, as those will turn out to be equivalent. For analysis, we assume that the activation function $\sigma$ is the identity (i.e. the system is linear), and that the population vectors take the equilibrium values of the above dynamics, that is

$$p^1 = (1 - \gamma_1)(\text{Id} - \gamma_1 W^r)^{-1}\phi^1 \tag{8}$$

$$p^2 = \gamma_2 W^f p^1 + (1 - \gamma_2)\phi^2. \tag{9}$$

We then define a learning rule for the synaptic weights, using these equilibrium values. Hence, we implicitly assume that neural dynamics happen on a timescale $\tau_p$ which is way quicker than that of learning, $\tau_W$. Taking the equilibrium values is the limit case which simplifies analysis, but in practice one can also take $\tau_p \ll \tau_W$ and simply update activities and weights concurrently. We also assume that our weight matrices are initialized in such a way that these are stable equilibria of the dynamics, which for example will be the case if all weights are initialized to sufficiently small non-negative values.

The learning rule we use is a slight modification of the learning rule used in [25]. In particular, we use the same general learning rule for both recurrent and feedforward weights, only varying certain parameters. Let $p^{post},i, p^{pre},i$ be the activity of the $i$-th post/pre-synaptic neuron respectively. We then update the weight from the $j$–th presynaptic neuron to the $i$-th postsynaptic neuron via

$$\Delta W_{ij} = \alpha \left(p_{t+1}^{post,i} - \sum_k W_{ik} p_t^{pre,k}\right) p_t^{pre,j} + \beta \left(p_t^{post,i} - \sum_k W_{ik} p_{t+1}^{pre,k}\right) p_{t+1}^{pre,j}. \tag{10}$$

The update rule contains terms of the form $p^{post},i p^{pre},j$, which are simply Hebbian terms. The other summands perform a subtractive normalization: they subtract the total overall input to a post-synaptic neuron, such that only activity exceeding this input will actually be considered positive. This term has been interpreted as a decorrelative term by [25] - in total, the learning rule can be understand as a predictive coding rule approximating a conditional expectation operation, as we explain in Appendix A in S1 Appendices. In matrix-vector notation the update rule reads

$$\Delta W = \alpha \left(p_{t+1}^{post} - W p_t^{pre}\right) p_t^{pre\,T} + \beta \left(p_t^{post} - W p_{t+1}^{pre}\right) p_{t+1}^{pre\,T}. \tag{11}$$

The parameters $\alpha, \beta \in \mathbb{R}$ control the sensitivity of this update to the order of the activity, put differently, the temporal symmetry of the rule: If we write our synaptic weight change as $\Delta_W(W, p_{t+1}^{post}, p_t^{post}, p_{t+1}^{pre}, p_t^{pre})$, then for parameters $\alpha = \beta$ we obtain a learning rule that is invariant under a reversal of time, that is

$$\Delta_W(W, p', p, q', q) = \Delta_W(W, p, p', q, q'), \tag{12}$$

while for the original learning rule with $\alpha = 1, \beta = 0$, no such relation holds - see also Fig 1. For $\alpha = -\beta$ we would obtain a rule that is antisymmetric in this sense - it turns out however that this would yield unstable dynamics.

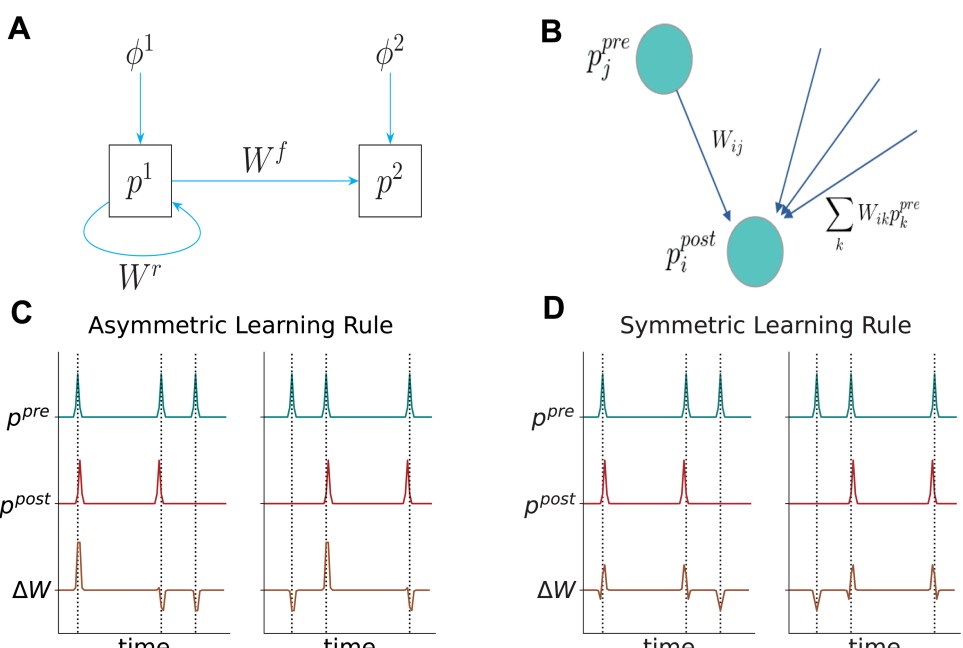

**Fig 1. Model and learning rule.** (A) Cartoon depiction of the model we are using in the main text. A recurrently connected population of neurons $p_1$, putatively $CA3$, receives external input $\phi_1$, putatively from dentate gyrus or entorhinal cortex($EC$). It projects to another population $p_2$, which receives input $\phi_2$. The latter could be $CA1$ receiving input again from $EC$. Note that there are no recurrent connections in the second layer and no backwards connections. (B) Quantities relevant for the update of synapse $W_{ij}$: pre- and postsynaptic activities, as well as the sum of the total input to the postsynaptic neuron through the synapses $W$. (C), (D) Invariance of learning rules with respect to temporal order. We plot synaptic weight change of a single synapse in a setup with a single pre- and postsynaptic neuron, respectively. The right column has the same pre- and postsynaptic activities as the left column, only in reverse order. In (C), the learning rule with parameters $\alpha = 1, \beta = 0$ is used, while in (D) $\alpha = \beta = \frac{1}{2}$. Only in the latter the synaptic weight changes are preserved (in reverse order), while in (C), postsynaptic activity before presynaptic activity leads to a net weight decrease. Note that in this illustrative example $W$ is fixed, in reality, network dynamics and weights would influence each other and lead to more complex changes.

## 2.3. Network learns successor representation and successor features

Before exhibiting the representations that are learned under the modified learning rule, it might be helpful understanding which representations a two layer-network as the above learns with the simplest choice of parameters ($\alpha = 1, \beta = 0$). Let us assume the network has been exposed extensively to features $\phi^1, \phi^2$ under the same random walk with transition probabilities $P$, such that the synaptic weights could converge. In practice this means simulating discretized dynamics of (Eq 6) together with the learning rule of (Eq 10), which then results in updates of the form ($\varepsilon_p >> \varepsilon_W$):

$$p^1(t+1) = p^1(t) + \varepsilon_p \Delta p^1(p^1(t), W^r(t), \phi^1(S_t)) \tag{13}$$
$$p^2(t+1) = p^2(t) + \varepsilon_p \Delta p^2(p^1(t), p^2(t), W^f(t), \phi^2(S_t))$$
$$W^r(t+1) = W^r(t) + \varepsilon_W \Delta W^r(p^1(t), p^1(t+1), W^r(t), \alpha^r, \beta^r)$$
$$W^f(t+1) = W^f(t) + \varepsilon_W \Delta W^f(p^1(t), p^1(t+1), p^2(t), p^2(t+1), W^f(t), \alpha^f, \beta^f)$$

$$\Delta p^1(p^1, W^r, \phi^1) = -p^1 + \sigma(\gamma_1 W^r p^1 + (1 - \gamma_1)\phi^1) \tag{14}$$

$$\Delta p^2(p^1, p^2, W^f, \phi^2) = -p^2 + \sigma(\gamma_2 W^f p^1 + (1 - \gamma_1)\phi^2)$$

$$\Delta W^r(p^1, p^{1'}, W^r, \alpha^r, \beta^r) = \alpha^r(p^{1'} - W^r p^1)p^{1^T} + \beta^r(p^1 - W^r p^{1'})p^{1'^T}$$

$$\Delta W^f(p^1, p^{1'}, p^2, p^{2'}, W^f, \alpha^f, \beta^f) = \alpha^f(p^{2'} - W^f p^1)p^{1^T} + \beta^r(p^2 - W^f p^{1'})p^{1'^T}$$

Then, as we show in Appendix A and Appendix D in S1 Appendices, the synaptic weights converge in such a way that the network computes successor features. Indeed, the equilibrium is best explained by stating what the population activities encode once the weights have converged. We find that the activities in the network, after extensive exposure to features $\phi^1, \phi^2$, when presented with new inputs $\tilde{\phi}^1, \tilde{\phi}^2$, compute successor representations/features: They take the equilibrium values

$$p^1 = (1 - \gamma_1)SF_{\phi^1}(\tilde{\phi}^1) = (1 - \gamma_1)\sum_{k=0}^{\infty}\gamma_1^k\mathbb{E}[\phi^1(S_{t+k})|\phi^1(S_t) = \tilde{\phi}^1] \tag{15}$$

$$p^2 = (1 - \gamma_2)\left(\tilde{\phi}^2 + \sum_{k=0}^{\infty}\gamma_2^k\mathbb{E}[\phi^2(S_{t+k})|SF_{\phi^1}(S_t) = SF_{\phi^1}(\tilde{\phi}^1)]\right). \tag{16}$$

In words, this means that the first, recurrent layer computes the weighted cumulative sum of the predicted values of the feature $\phi^1$ it was trained on, but given the possibly new feature $\tilde{\phi}^1$ - one could see this a form of pattern completion, but with a predictive component. Similarly, the second layer computes predictions of $\phi^2$, only that these predictions themselves depend on those predictive representations passed to it from the first layer. In particular, when the inputs are the same as the model was trained on, and the maps $\phi_i$ are injective (i.e., sufficiently rich features exist for the environment), then these equations simplify to

$$p_1 = (1 - \gamma_1)SR_{\phi_1} \tag{17}$$

$$p_2 = (1 - \gamma_2)SR_{\phi_2}. \tag{18}$$

That is, in this case the two layers simply learn to encode the successor representations of their respective inputs.

## 2.4. Influence on the representations by choice of parameters

We now proceed to ask the question "which representations would be learned depending on the choice of the parameters $\alpha, \beta$?". It turns out that the resulting representations are still successor representations, albeit corresponding to transition probabilities that are not necessarily faithful to those of the environmental dynamics anymore. To be precise, we define a weighted sum of transition matrices

$$P_{\alpha,\beta} := \frac{\alpha}{\alpha + \beta}P^{forward} + \frac{\beta}{\alpha + \beta}P^{backward}. \tag{19}$$

Here, $P^{forward}$ contains the transition probabilities of the actually observed process (that is, 'forward' in time), while $P^{backward}$ contains the transition probabilities of the reverse process, i.e. $p(s_t = s|s_{t+1} = s')$ - see also (Eq 39) We then show in Appendix D in S1 Appendices that

under suitable conditions, with the learning rule defined above, the model is able to learn the successor representations under $P_{\alpha,\beta}$. This entails that the neuronal population activities at convergence yield

$$p_t^i = SR_{\phi^i}^{P_{\alpha^i,\beta^i}}(\phi_t^i) \tag{20}$$

with $\alpha^i, \beta^i$ the respective parameters used in the learning rule. In terms of the successor representation matrix, this simply means

$$SR^{P_{\alpha,\beta}} = \sum_{k=0}^{\infty} \gamma^k (P_{\alpha,\beta})^k. \tag{21}$$

That is, in particular, under the regime $\alpha = 1, \beta = 0$, this corresponds to the 'true' successor representation. For $\beta = 1, \alpha = 0$ one obtains the 'predecessor representation' [16]. Under a symmetric regime, the transition probabilities forward and backward in time are averaged over, that is one obtains

$$P_{\frac{1}{2},\frac{1}{2}} = \frac{1}{2}(P^{forward} + P^{backward}). \tag{22}$$

In fact, in this case the transition probabilities are *reversible* in time - this is not surprising, as the learning rule was defined to be invariant under a time reversal and hence should only extract aspects of the dynamics which are reversible. We note here that such reversible dynamics have been typically assumed in the theory of SR when construing grid cells as efficient representations of the geometry of an environment through the eigenvectors of the SR - see Appendix C in S1 Appendices.

**2.4.1. Activities in the model converge to theoretically obtained limits.** Having theoretically obtained the limits of the weights and the corresponding activities, we next verified these limits empirically in simulations. In these simulations, we consider an environment with a discrete number of states $s \in \mathcal{S}$ and inputs $\phi^i$ which are functions of these states. In the simplest case, $\phi(s) = e_s$, we obtain the classical successor representation. In particular, for a symmetric learning rule we obtain a symmetrized successor representation - which shows less dependence on the policy. For example, on a circular track, the representation becomes indifferent to whether the agent is performing a clock-wise or a anti-clockwise walk (Fig 2). One might argue that a reversible representation encodes more of the geometry of the underlying state space and less of the actual dynamics (although there is still an indirect influence of the dynamics through the stationary distribution). Indeed, we show in Appendix G in S1 Appendices that the symmetrized transition probabilities are always closer to a uniform policy than the unsymmetrized ones.

We also verified our theoretical results in more complex scenarios: the convergence prevails also when the features are random inputs instead of one-hot vectors, and also when the random walk is arbitrary instead of circular - see Fig 3. Additionally, we investigated the stability under the choice of parameters $\alpha, \beta$. Here we found that it seems that the model is only stable when the positive weight is bigger (in absolute value) than the negative weight, with no convergence at the boundary case of $\alpha = -\beta$. The latter is in line with the theoretical results - note that (Eq 19) is undefined in this case.

## 2.5. Place fields under symmetrized rule show less shift

Although both areas CA3 and CA1 show place cells, these cells exhibit different properties and dynamics [33,34]. It is well known, that on a linear or circular track place fields shift

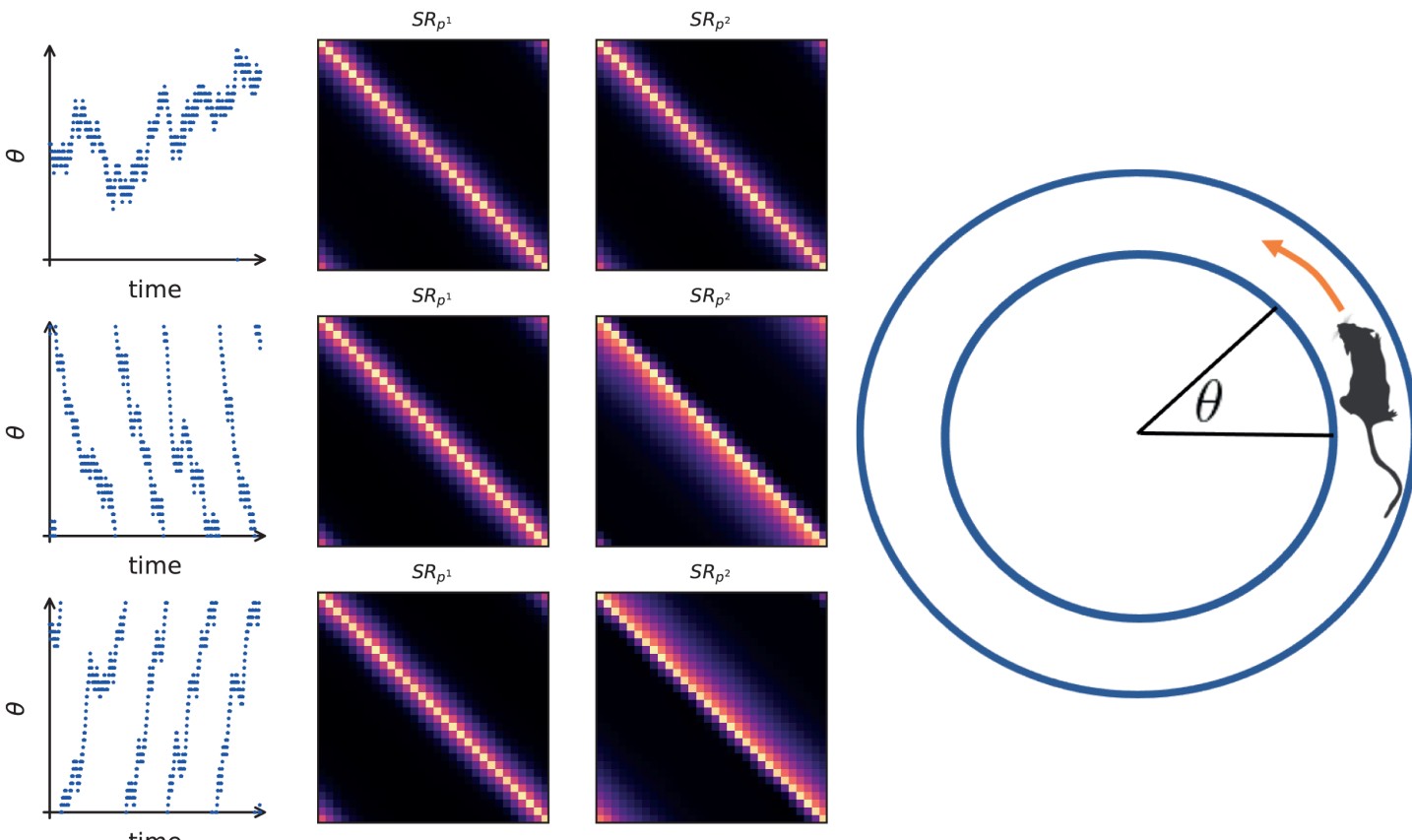

**Fig 2. Successor Representations learned in circular random walks.** We construct a circular state space with possible actions stay, move clockwise and move anti-clockwise. We simulate three random walks, one where the actions are selected uniformly (first row), one where clockwise actions are preferably selected (second row) and one where anti-clockwise actions are preferably selected (third row). The first column shows an example trajectory of the respective walk. The second and third column show the successor representations learned by the first and second layer of our model, using a symmetric ($\alpha = \beta = \frac{1}{2}$) and an asymmetric ($\alpha = 1, \beta = 0$) learning rule, respectively. Note how the successor representation learned with a symmetric rule does not distinguish between the policies. Here, the inputs to the cells are one-hot vectors encoding the respective states and the plotted successor representations are obtained by taking the average population activity in the respective states.

backwards opposite the direction of travel in both regions [35,36]. However, when directly comparing cell recordings from both, it has been observed for example in [37] that the shift in CA3 is in general less pronounced, that is, the center of mass of these cells is more stable than in their counterparts in CA1. We hypothesized that a difference in learning rules could explain this effect. Indeed, it is not hard to see theoretically why this should be the case: Through learning, the features $\phi^i(s)$ get replaced by their successor features $SF_{\phi^i}(s)$. If there is a preferred direction of travel, then states preceding those where the feature puts a lot of mass will also have more mass, since they are predictive of the former states. If there is an asymmetry in the policy, the same will not hold true for succeeding states, hence one observes a shift towards the predecessors. If now however one has symmetric transition probabilities, then there is no directionality, hence this shift would not occur. Indeed, we provide a simple proof of this in Sect 4.5.

We confirmed this intuition, running our two-layer model in a simple linear track where the agent repeatedly moves from the left side to the right. Indeed, we find a tendency to shift in the CA1 cells, which isn't as pronounced in the CA3 population (Fig 4). Qualitatively, our

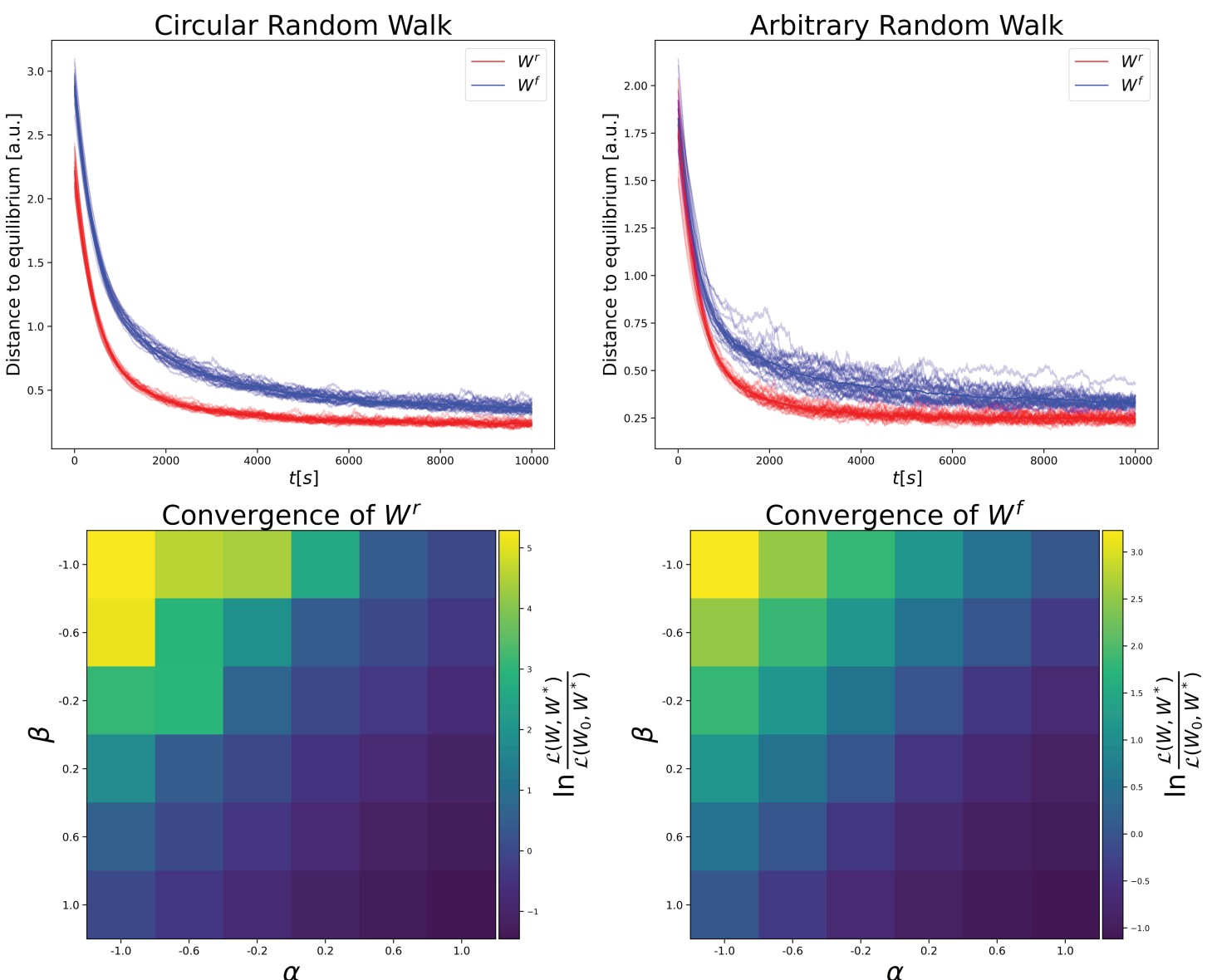

**Fig 3. Successor representations are learned for a variety of inputs, dynamics, and parameters. Top:** Convergence of recurrent (red) and feedforward (blue) matrices to their theoretical limit with random features in circular (**left**) and arbitrary (**right**) random walks. **Bottom:** Convergence of recurrent weight (**left**) and feedforward weight (**right**) for different parameters $\alpha, \beta$. The other set of parameters is fixed to $(1,0)$ and $(\frac{1}{2}, \frac{1}{2})$ in these experiments, respectively. In graphs, we measure convergence by the loss term $\mathcal{L}$ as explained in Methods section. In the bottom row, we compute the fraction of the loss at the final step over the initial loss and display the result in a logscale. Thus, negative values indicate converging towards the target. Note that the values on the antidiagonal are approximately 0.

results match those obtained in [37]. Importantly, these results only hold when using the symmetrized version of the learning rule for CA3, while the asymmetric variant yields almost no distinction.

## 2.6. Generalization and learning with (a)-symmetric rules

Having derived the different kinds of successor representations that symmetric and asymmetric learning rules encode, we next sought to understand what the functional benefits of these

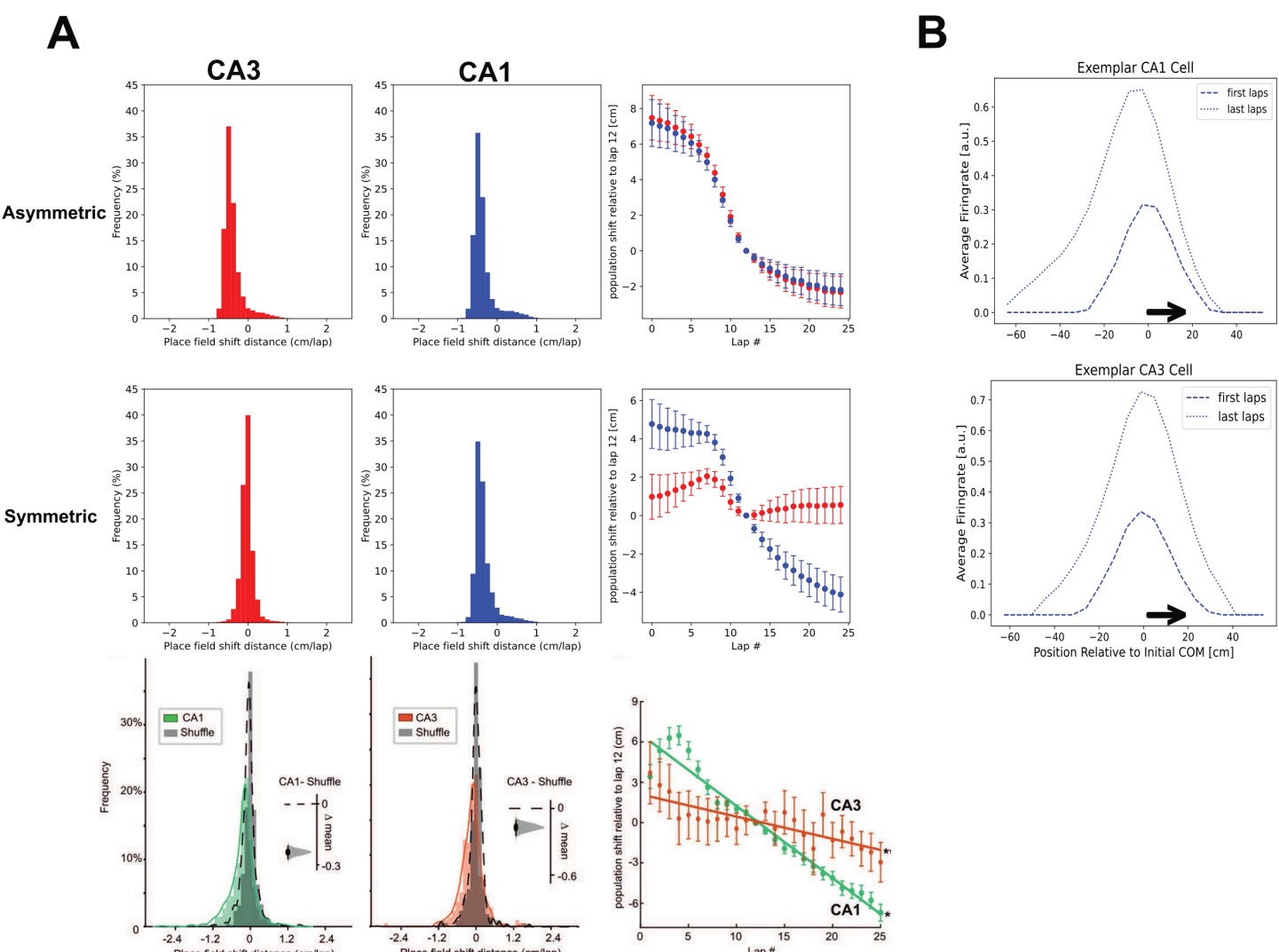

**Fig 4. Symmetric learning rule leads to more stable place fields in linear track.** We simulated an experiment with a rat repeatedly running on a linear track, similar to [37]. A two-layer SR network was used where the recurrent weights had a symmetric (**A, middle row**) or asymmetric (**A, top row**) learning rule. In the the symmetric case, there is less shift of the centre of mass of place fields in the modelled CA3 population (*red*) than in the CA1 population (*blue*), which is not the case in the asymmetric version. Histograms show distribution of shifts comparing last five laps versus first five laps, while the rightmost plot shows shift relative to the 12-th lap. The results in the symmetric case are qualitatively similar to data (**A, bottom row**) from $Ca^{2+}$ recordings of hippocampal neurons in a similar experiment - figure adapted from Dong, C., Madar, A. D., & Sheffield, M. E. (2021)( [37]). In **B**, we show firing rates of an exemplar cell from CA3 and CA1 respectively, where the symmetric learning rule is used for CA3. The firing rates in each position are averaged over the first and last five laps, and plotted relative to the centre of mass in the first laps. With experience, only the place field in CA1, not the place field in CA3 shifts backwards (arrow indicates direction of travel).

representations might be. In particular, we hypothesized that a symmetric learning rule for successor representations might be a relatively simple inductive bias that would favour learning such representations that are invariant under time reversal. This could be useful in such environments where there is a symmetry in transition structure. A particularly simple example of this setting - but still likely for biological agents to encounter - is when the transition structure of the environment is deterministic and transitions in both directions between states are possible. In this case, the state space becomes a metric space and the metric a particular invariant under the symmetry - that is $d(s, s') = d(s', s)$. This then hints at a possible benefit of

using a learning rule biased towards such symmetry: In a metric space, an optimal policy for navigating towards any target depends only on the metric - in fact one may give a closed form expression as we show in Appendix H in S1 Appendices. Hence, the more the representations that are learned in one particular tasks encode something akin to the distance on the underlying space, the more useful these representations should be for generalizing to new such tasks. In other words, one would want to bias the representations to encode more of the geometry of the space and less of the dynamics of the particular tasks.

In the light of this hypothesis, we trained a reinforcement learning agent, equipped with a temporal difference (TD) learning rule that for a fixed policy would converge to the same weighted representation under $P_{\alpha,\beta}$, and investigated performance in simple navigation tasks. Note that since we now want to understand the benefits on the computational level irrespective of the biological implementation, we use a classical RL model and not the neural network model from the preceding sections - however, all experiments could also be conducted using such a model. The agent we use encounters transitions of states and updates an internal matrix $M$ which serves as an estimate of the successor representation. When transitioning from state $s \to s'$, the update equations are given by

$$M(t + 1) = M(t) + \varepsilon \Delta M \tag{23}$$
$$\Delta M_{u,v} = \alpha \delta(u = s)\left(\delta(v = s) + \gamma M_{s',v} - M_{s,v}\right) \tag{24}$$
$$+ \beta \delta(u = s')\left(\delta(v = s') + \gamma M_{s,v} - M_{s',v}\right)$$

where $\varepsilon$ is a fixed learning rate. Note that for $\alpha = 1, \beta = 0$ this yields the standard TD learning rule for the successor representation [14]. Furthermore, the agent learns a reward vector $R$, and computes the value function of states via $V_R(s) = MR(s)$. Together with a local transition model $p(s'|s, a)$ (which we assume as given), the agent can then define $Q$ values of state action pairs $Q(s,a)$ and take the next action based on these $Q$–values [11,38].

The task for the agent is split in two parts: In the first part, in each epoch, the agent is initialized in a random location and has to navigate to a fixed goal $s_{target}$, where a unit reward is received -i.e., $\phi = e_{s_{target}}$. This goal does not change over epochs. After a fixed number of epochs, the goal is changed to a new location $s'_{target}$, randomly drawn from all other locations. Importantly, the agent is then only allowed to relearn the reward vector, not the successor representation matrix.

We find that both the classical SR, corresponding to parameters $\alpha = 1, \beta = 0$, as well as the symmetrized version $\alpha = \beta = \frac{1}{2}$ are able to learn the navigation tasks with similar mean learning curves. Although it might be counterintuitive at first that symmetrization yields a policy which still navigates to the correct target, we actually prove in Appendix H in S1 Appendices that indeed an optimal policy is stable under such symmetrization - which shows at least that once such a policy is learned, it can be maintained. However, this result only shows stability once an optimal policy is reached, and indeed one may observe in Fig 5 that the symmetric agent generally shows a higher variation. Furthermore, we find that the symmetric agent seems to be more sensitive to the choice of hyperparameters: Indeed, we find that for smaller learning rates, there is a steeper degradation of performance caused by high variability in the results for the symmetric agent. However, for higher learning rates, both show a similar performance (Fig 6).

Importantly, we then find empirically that on the new targets, the symmetric learning rule outperforms the classical one on average, while both show higher variation in these tests (Fig 5). Thus, one may argue that the successor representation in a symmetric learning regime affords better generalization - at least in a navigational setting. This is not merely an effect

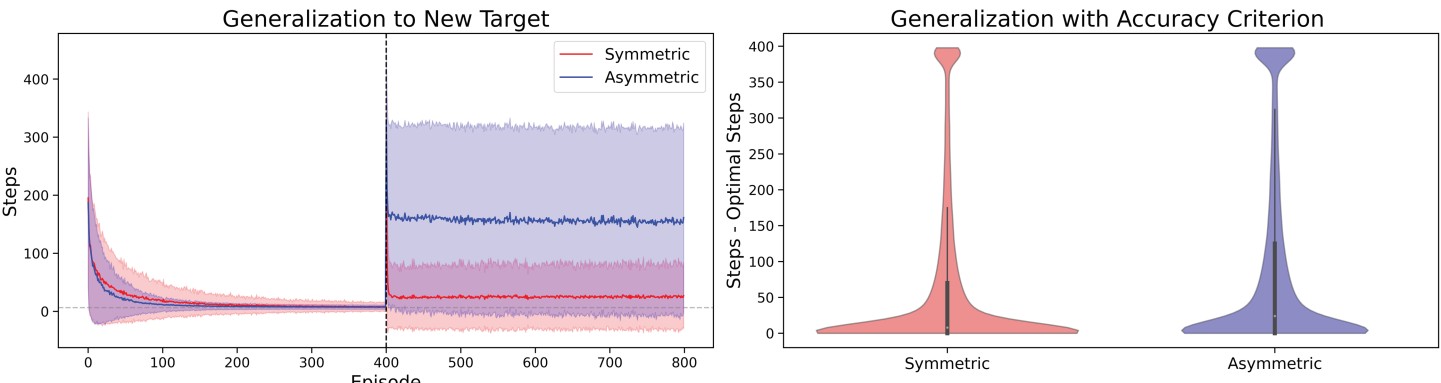

**Fig 5. Symmetric successor representation agent affords better generalization in simple navigation tasks**. **Left:** Agents started in random locations in the environments and had to learn to navigate to fixed targets. After 400 episodes, reward location was switched to a new random location, where agents could only relearn the reward prediction vector but not the SR. (Generalization) performance is visualized by total number of steps taken per episode, for an agent using the classical rule (blue) and an agent using the symmetric rule (red). Dashed line indicates change of target location. We show the average performance over different environments as performance is qualitatively similar, see S3 Fig for plots in individual environments. **Right:** Similar to left plot, but instead of switching target after a fixed number of episodes, the target was switched when the previous target was found with a fixed accuracy. Violin plots show distribution of suboptimality (steps - optimal number of steps) over all environments, for individual environments see S4 Fig. For an outline of the environments see S2 Fig.

of the trajectories that are sampled with the different learning rules - that is, in particular it cannot be attributed to the higher variation during training on the first target: We repeated the above experiment while learning the successor representations based on the transitions obtained from the classical agent alone - the results remain unchanged, suggesting that the symmetric rule yields representations more apt to generalization without the need of a different sampling regime (S5 Fig). Similarly, the results also hold when controlling the norm of the updates, such that both the asymmetric and symmetric update make an equally big update step at each point. In other words, the agent can concentrate on solving the current task and still gets afforded a map of the environment which is less influenced by the current policy. Nevertheless, when learning to navigate to the first target, the symmetric agent still learns a policy that on average has more entropy than the one learned by the asymmetric agent. This increase in entropy then yields a generalization advantage: as soon as the new target is introduced, the effect is reversed and the asymmetric agent has the more entropic policy (Fig 6).

Indeed, one can show that the transition probabilities encoded in $P_{\frac{1}{2},\frac{1}{2}}$ will be closer (than the observed transition probabilities) to those that correspond to a uniform policy, choosing every transition with equal probability - see Appendix G in S1 Appendices - in particular meaning they have higher entropy. The successor representation of the uniform policy in turn is closely related to the shortest-path distance [39]. It can thus be used to generalize to *any* navigational target, while the successor representation under other policies will not neccesarily have this property. Together with our previous considerations on the symmetry of the state space, this led us to hypothesize that the generalization effect of the symmetric learning rule should vanish as soon as there is no such symmetry in the state space any more. We thus repeated the above experiment on a state space that corresponds to a directed graph, where the number of transitions needed to go from $s$ to $s'$ is not necessarily equal to those needed to travel from $s'$ to $s$. Indeed, we find that in such a setting the effect is reversed: there, the classical learning rule leads to better generalization (Fig 7).

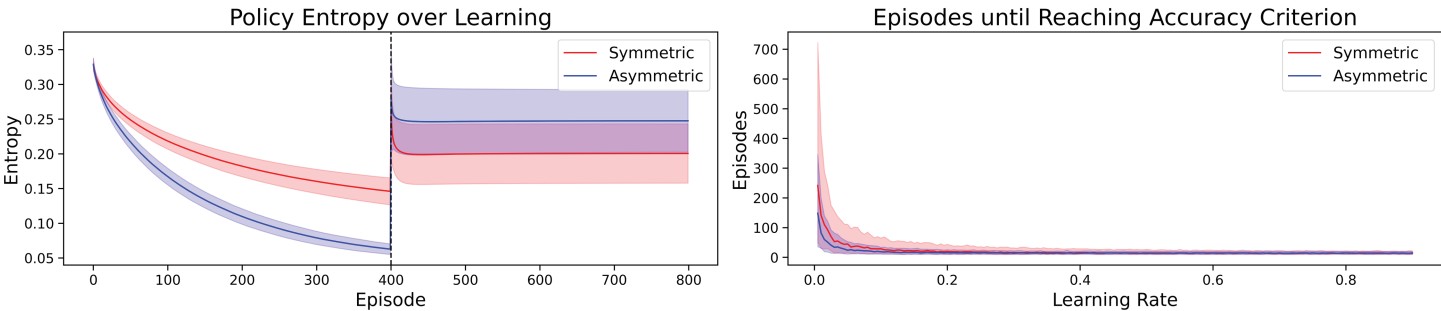

**Fig 6. Comparison of policy entropy and sensitivity to learning rate among agents with symmetric and asymmetric learning rule. Left:** Policy entropy of the two agents show different trajectories in the generalization experiment. We calculated the entropy of the agent's policy, averaged over all states, at the end of each episode. This reveals that during learning to navigate to the first target, the symmetric agent has more entropy, which is then reversed when the new target has to be reached. **Right:** Symmetric agent shows more sensitivity to learning rate parameter for lower learning rates. We trained the agents repeatedly until a fixed accuracy in navigation to the target was met. We then recorded the number of episodes it took until that criterion was reached. Curves show median and interquartile range of this number for the two agents.

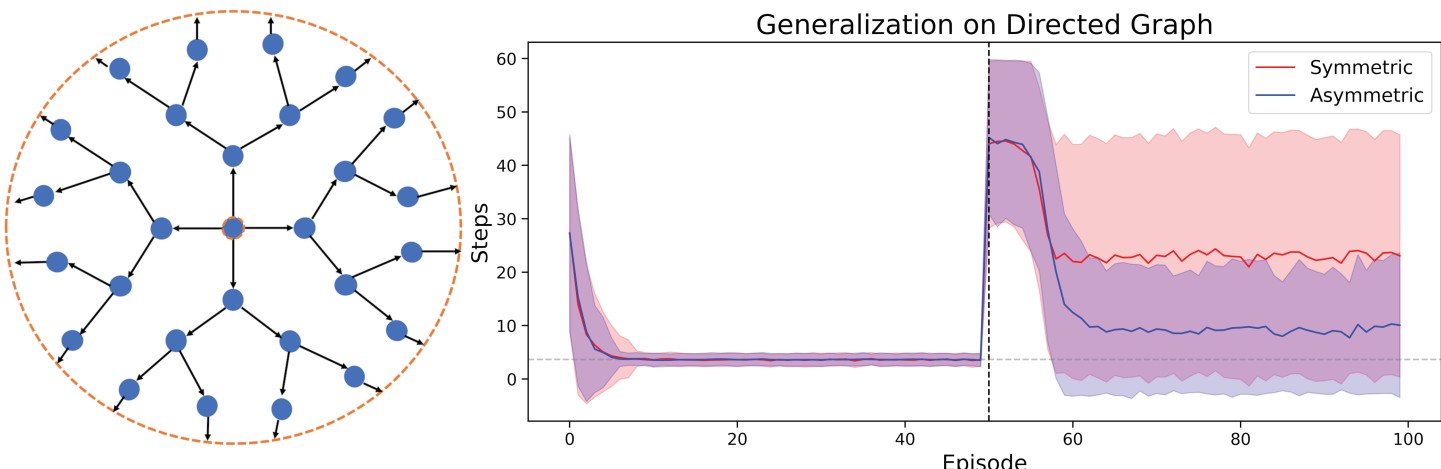

**Fig 7. Symmetric learning rule provides no advantage in generalization experiment on a directed graph.** We conducted the same kind of experiment as in Fig 5 on a directed graph. **Left**: The state space is tree-like, with the addition that from the leaf nodes at the last level one travels back to the central node (orange dashed line). **Right**: In this scenario an SR agent with the classical learning rule (blue) performs better in generalization than one with the symmetric learning rule (red).

## 2.7. Variation in symmetry

So far the temporal difference learning rules we used had either perfect symmetry or no symmetry at all. In biological agents, such perfect symmetry might rarely be given, rather, one might expect that the temporal sensitivity profile of learning rules could vary both in space and time, meaning that cells exhibit different learning rules at different moments and furthermore two cells might vary in how they learn. We wondered how robust our findings are to such variations. Thus, we investigated the generalization performance of our successor representation agents under variations that might correspond to imperfect symmetry. First, we investigated how well agents would generalize that have a learning rule in between the symmetric and the classical ones we have discussed so far. To do so, we chose parameters $\alpha = \frac{1}{1+s}, \beta = \frac{s}{1+s}$, where $s$ is a parameter that we varied. Note that for $s = 0$ we get the classical rule, while for $s = 1$ we get the symmetric rule. We found that generalization performance

increased with the parameter $s$, with the highest generalization at $s = 1$, again confirming that the symmetric rule might afford a representation that generalizes best.

To further assess the robustness of our results, we then investigated whether dropping the assumption of globally fixed values $\alpha, \beta$ would qualitatively alter the results. Again, this is motivated by the assumption that in biological agents, learning rules will not be perfectly static but might vary.

To capture this in the RL setting, first, we introduced noise to the parameters at every time step. That is, our update rule for the successor representation would become

$$\Delta M_{u,v}(t) = (\alpha + \nu_\alpha(t))\delta(u = s)\left(\delta(v = s) + \gamma M_{s',v} - M_{s,v}\right) \tag{25}$$
$$+ (\beta + \nu_\beta(t))\delta(u = s')\left(\delta(v = s') + \gamma M_{s,v} - M_{s',v}\right)$$

where $\nu_\alpha, \nu_\beta$ is independent noise that is added at each timestep to the parameters. Secondly, we studied a condition where we introduced a state-dependent heterogeneity in the parameters. That is, for each possible transition $s, s'$ in the state space, there is a separate set of parameters $\alpha(s, s'), \beta(s, s')$ which are randomly initialized at the start of learning and then fixed. The update rule thus is

$$\Delta M_{u,v}(t) = \alpha(u, v)\delta(u = s)\left(\delta(v = s) + \gamma M_{s',v} - M_{s,v}\right) \tag{26}$$
$$+ \beta(u, v)\delta(u = s')\left(\delta(v = s') + \gamma M_{s,v} - M_{s',v}\right).$$

To determine the parameters, first some putative values $\tilde{\alpha}, \tilde{\beta}$ are drawn from Gaussian distributions with identical variance and different means $\mathcal{N}(\mu_\alpha, \sigma), \mathcal{N}(\mu_\beta, \sigma)$, and afterwards the absolute value of those is taken. This is to ensure that we don't have negative parameters, which might result in qualitatively different and unstable learning rules. The means $\mu_\alpha, \mu_\beta$ are then either set to $(1,0)$ to mimic the asymmetric with inhomogeneous parameters, or to $(\frac{1}{2}, \frac{1}{2})$ for the symmetric agent. In both variations we again find qualitatively similar results as before: even with noisy parameters, both agents are able to learn simple navigation tasks (Fig 8). The symmetric agent shows more variation in performance, but is in turn able to generalize better to new targets.

## 2.8. Maze tasks

The grid worlds we used to investigate the generalization capability so far were topologically very simple. In reality, biological agents will be exposed to more complicated navigation situations, with multiple paths and detours. We thus repeated our setup in a more contrived maze with multiple chambers and more convoluted paths. In particular, instead of keeping the environment static, when we tested the generalization abilities of the agents we introduced slight modifications, such that previously open paths were blocked. Again, in this setting the symmetric agent showed a better generalization, indicated by a higher probabilty of solving the new task with a number of steps close to the optimal one, as indicated by the distribution plots in Fig 9. However, in comparison to simpler grid worlds, both agents are worse at generalization in this setting, which might be caused by the additional complexity of the environment.

## 3. Discussion

Here, we have expanded the existing work on successor representation models of the hippocampus. We extended previous local learning rule models by including learning at two

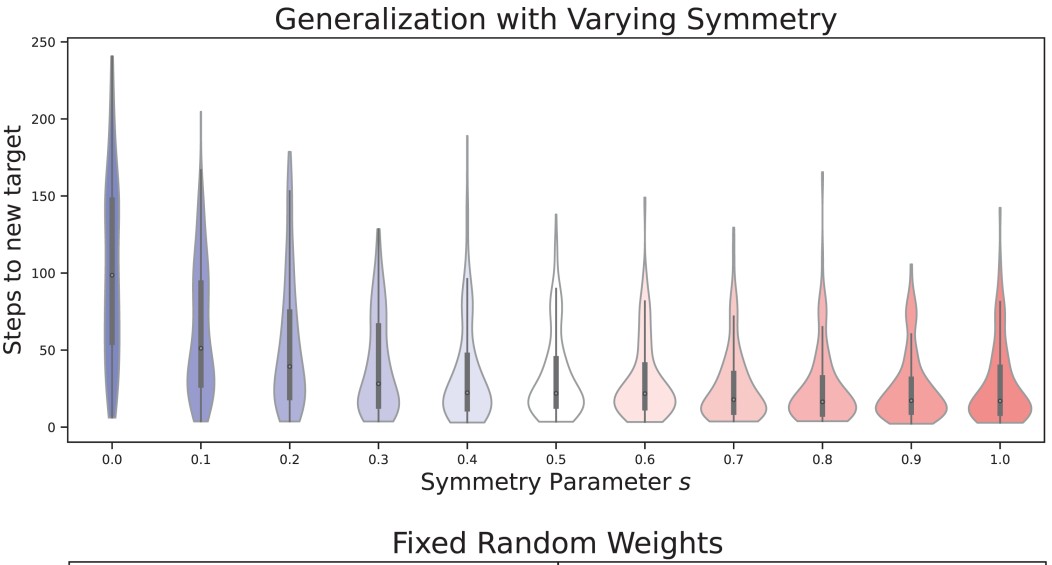

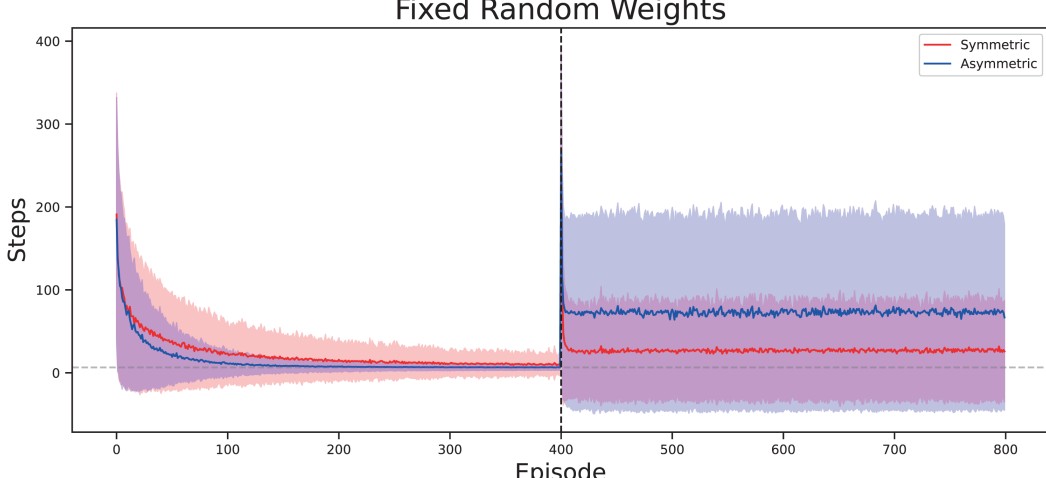

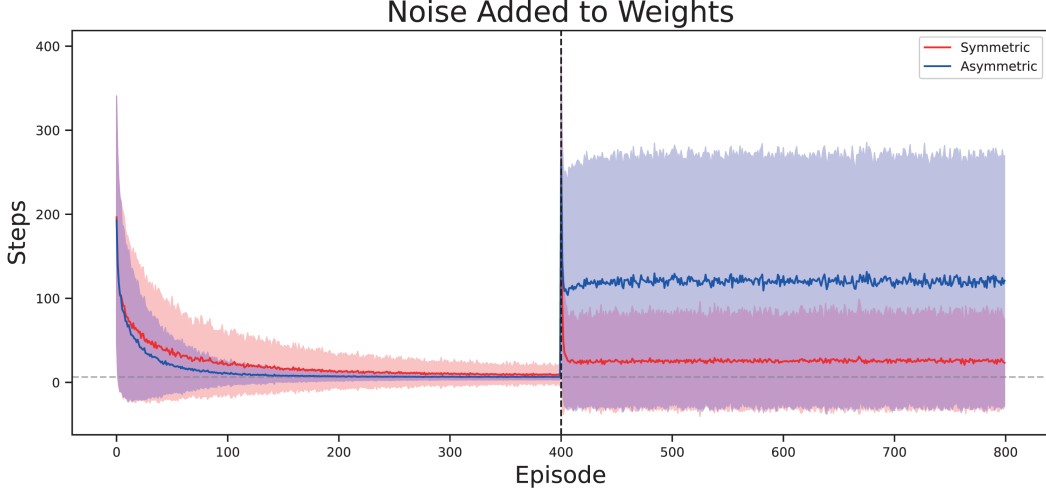

**Fig 8. Variations of symmetry in the learning rule** Experiment for all plots is the same as in Fig 5. **Top:** Generalization for parameters $\alpha = \frac{1}{1+s}, \beta = \frac{s}{1+s}$. Violin plots show distribution of differences (steps-optimal number of steps) when evaluated on new targets. Distributions broaden towards the optimal value of 0 with increased symmetry. **Middle:** Generalization with parameters $\alpha, \beta$ randomly initialized for each pair of states. **Bottom:** Generalization with noise added to $\alpha$, $\beta$ at each timestep.

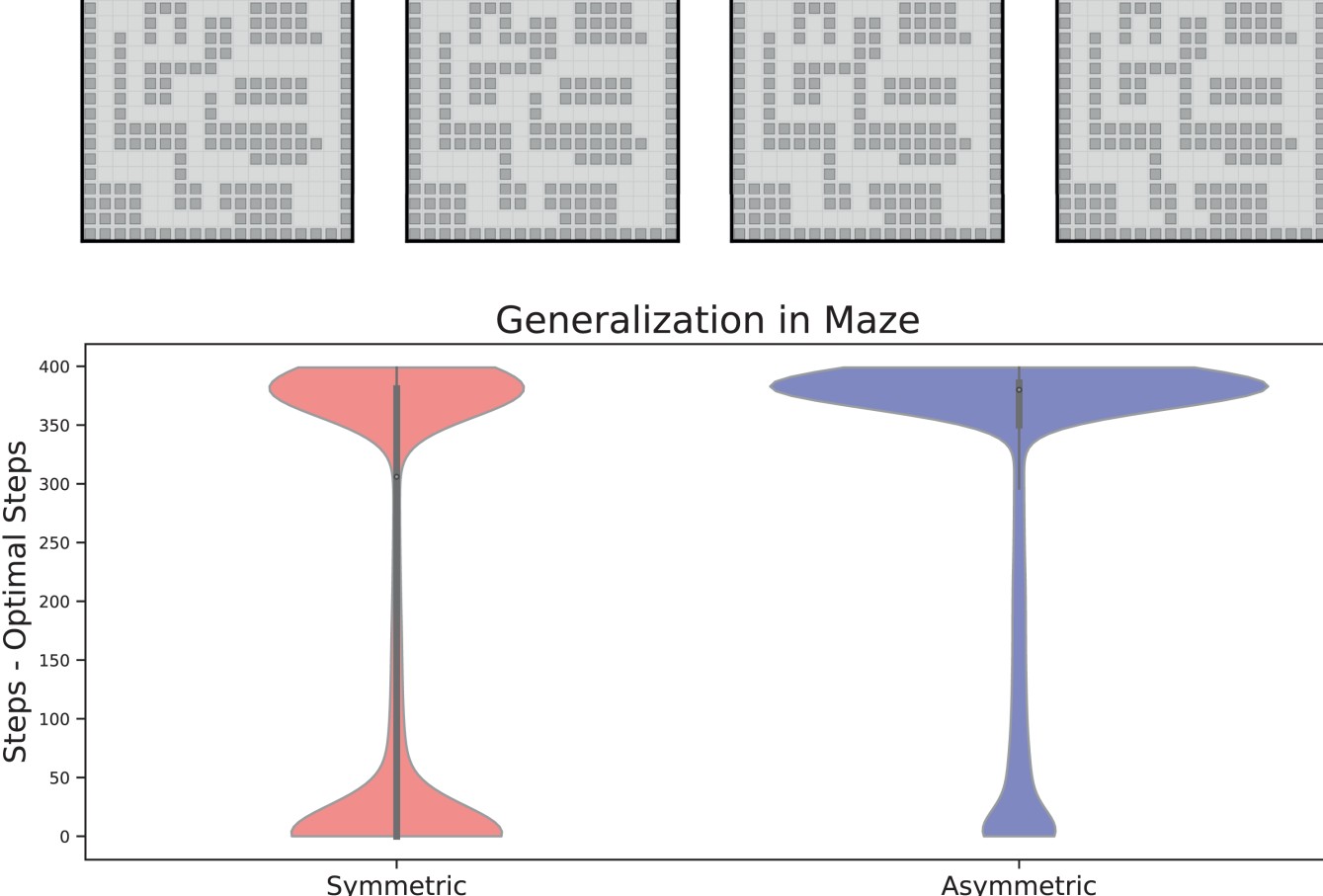

**Fig 9. Generalization performance in maze task with blocked paths Top**: Grid world mazes used for generalization task. Leftmost maze was used for training, the other three environments for testing the generalization. **Bottom**: Violin plots show distribution of suboptimality (steps - optimal number of steps) of the agents when using the successor representation trained on one target and tested on another one. Training and test targets are randomly drawn from all possible states in the respective environments. The distribution for the symmetric agent is broader around 0, which indicates optimal generalization, and less pronounced at 400, which was the maximum number of steps allowed.

synapses, corresponding to CA3-CA3 recurrent synapses and CA3-CA1 feedforward synapses. In this model, we then studied the effect of making a local learning rule invariant to a reversal of time on the learned representations. We have found that the successor representations that are learned under such a learning rule correspond to encoding a transition structure which is also invariant to a time reversal - irrespective of the actual dynamics which are experienced. In particular, we could show that under such a symmetrized learning rule, place fields shift less when on a linear track, which is in line with empirical findings showing a distinction between place field shifting in CA3/CA1. Although we show that our model is able to replicate the observed shifting effects from [37], further experimental data would be needed to corroborate these findings. Indeed, albeit CA3 place fields are generally reported as being more stable [40], there are additional experimental and behavioural covariates to consider: For example, although [36] find that the centre of mass of place fields in CA1 shift while those in CA3, are relatively stable, this holds only for familiar environments. In novel environments, both exhibit shift, although more pronounced in CA1. This speaks to the fact that a static,

symmetric learning mechanism alone might be insufficient to account for all data, and plasticity might generally vary - both in strength as well as in symmetry. Additional experiments would be needed to disentangle such influences.

The local learning rule we have modified from [25] is not the only one which could be used to obtain successor representations. Indeed, also [41] and [24] use local learning rules to learn these representations. It seems plausible that to obtain the successor representations we studied here, the exact shape of the learning rule is not important, as long as one can symmetrize it in an appropriate way. This might not even necessarily mean symmetrizing the plasticity kernel: For example, [41] use STDP to learn feedforward connections between CA3 and CA1 neurons. Specifically, phase precession was used to provide a location code in the timescale of STDP, and importantly, in the absence of phase precession, a symmetric SR is learned in a biased random walk. Thus, it is plausible that precession in their model breaks the temporal invariance.

To understand the functional significance of such learning rules, we then went on to reinforcement learning experiments, where we trained an agent in navigation tasks. Here, we could show that a symmetrized learning rule affords a better generalization performance when the agent should navigate to a new target. This is interesting, because successor features have been explicitly employed in RL to obtain better generalization to new tasks [11,12,42]. In particular, also when the SR was introduced as a model for hippocampus in the neuroscience literature, the generalization capability of such representations was measured [8,14]. In fact, it was argued in [8] that especially a successor representation that corresponds to a uniform policy should be beneficial for generalization. This was later also identified as a flaw of classical successor representation theory when linear reinforcement learning was suggested as a model for the hippocampal formation instead [43]. In the latter, instead of storing the successor representation under the current policy, the representation under a default policy is stored, and the current policy can be efficiently represented by only encoding the deviations from default. This default policy in navigation is of course intuitively the uniform policy, which could be learned by an agent as soon as it encounters a new environment, in an explorative manner. Our symmetrized learning rule provides a middle ground between these two perspectives: the successor representation that one learns with this learning rule is depending on the current policy, but one can show that the reversible version is always closer to the SR of uniform policy than the SR under the original policy (see S1 Appendices). Thus, an agent does not necessarily have to start with exploration to construct a map of its environment, but can do so while performing a particular task. This is of course only possible due to the inductive bias that is inherent in the learning rule, which assumes that the state space is symmetric, since we observed that when this assumption is not true, the SR affords worse generalization.

Adapting learning mechanisms to symmetries of the data is a topic under ongoing investigation in biological and machine learning communities [44,45]. In reinforcement learning, symmetries are frequently considered under the framework of MDP homomorphisms [46–48]. This line of research for example aims at learning efficient abstractions of large state spaces into more amenable ones, or exploits known symmetries in the task structure to learn more efficiently. The symmetric state spaces we consider here are simple cases of an equivalence of states induced by the optimal value function: in a navigational problem in a metric space, any two states are equivalent with respect to the optimal value function if they have the same distance to the target [49]. It would be interesting to investigate whether modifications of TD learning are also beneficial for the more general case of symmetries that are considered in this literature. Interestingly, it has been shown before that TD learning can be considered as performing a form of gradient descent if and only if the dynamics under the policy are

reversible [50]. This is intriguing, since TD-learning is known to be unstable in the continuous setting [38] - if our symmetrized learning rule extends to this setting, then it might be possible that it could be useful for a more stable learning in symmetric settings.

Using a fixed learning rule for all synapses of a region might be a simplistic assumption, and in real networks, possibly a variety of learning rules are at play [51]. Our model of course does not fully capture this diversity, although we have investigated the stability of our RL model under noisy variations of the parameters. However, it would be easy to adapt the learning rules of individual neurons in such a sense that they have varying temporal profile. This could then possibly lead to a whole spectrum of successor features, each with its own sensitivity to future and past. In fact, it has been proposed before that the hippocampus encodes representation of both predecessors and successors [52], where representing preceding states has furthermore been suggested as useful for exploration in unsupervised reinforcement learning [16]. These purely predictive respectively postdictive representations would be the two ends of a continuum of representations, with the symmetric learning rule in the middle. Thus, a model representing such a continuum could provide a more nuanced understanding of neural encoding and learning processes

Furthermore, it should be noted that there is a picture emerging where synaptic learning mechanisms are not but static but rather are controlled by neuromodulators, which could determine whether a certain firing pattern in pre- and postsynaptic neurons will result in potentiation or depression [53–55]. Indeed, prominently (but not exclusively) dopamine and acetylcholine will modulate learning rules - both of which are important in reward-based learning and navigation [56,57]. In particular, it has been shown in a computational model that switching between plasticity regimes in a transmitter-dependent manner may result in successful spatial learning, in navigation tasks similar to the ones we considered here [58]. It may thus be an interesting avenue for future research to analyze such dynamically switching regimes in terms of successor representations to understand which predictive operation they compute. This might for example be achieved in the model presented here by modulating the parameters $\alpha, \beta$ depending on some external variable like reward.

Additionally, it is noteworthy that replay phenomena, where sequences of cells corresponding to recently visited states are reactivated in an orderly, time-compressed fashion, are taking place in hippocampus both in a forward as well as a reverse direction [59,60]. From the perspective of reinforcement learning, for offline planning agents might indeed want to be able to simulate replay in the backward as well as the forward direction [61,62]. These replay processes would provide yet another mechanism by which a symmetrized successor representation could be learned: Indeed, instead of modifying the TD learning rule during online learning, one could instead apply the classical learning rule during offline learning on replayed trajectories. This should again lead to a symmetrized representation, if forward and backward trajectories are replayed with equal probabilities. On the other hand, a symmetric learning rule in CA3 has been identified as a key mechanism to generate replay in the reverse direction [63].

Our model naturally is a broad oversimplification of matters and lacks biological realism. This is not a problem per se, because we aimed here at analytical amenability and to expand on the theory of successor representations, which operates on the computational level. Still, our focus was obtaining an explicit relation to successor representations by considering CA3/CA1 in isolation, with external input synapses not subject themselves to learning. In reality, it is now accepted that there is not a simple forward pass through the hippocampus, but rather there are projections from the deep layers of entorhinal cortex back into the superficial layers (which provide input to hippocampus), essentially creating a loop [64,65]. With

plasticity also happening at these synapses, one would then obtain a model that is not as simple anymore as the one we presented here. Studying the representations in such an extended model, which would include learning for example in synapses from HC to EC, and whether these can still be framed in Successor representation theory, would be an interesting future research direction and could possibly build a bridge to computational models which include hippocampal-entorhinal interactions [66,67]. Speculatively, when including spatially selective cell types like grid cells from the entorhinal cortex in the loop, the general result that HC represents successor features should not change. Indeed, taking the input from multiple grid cell modules together, these cells provide a unique encoding of spatial position [68]. This is thus an injective function of the current spatial state, just as we used as input for our system. These coordinates, as they encode space, naturally benefit from a symmetrized representation in CA3. One step further one could then include additional, non-spatial input externally to CA1. This would then result in a successor representation of the non-spatial features, conditioned on the spatial features. That is, a long-term prediction of expected stimuli, given current position in space. Connections back from hippocampus to EC might in turn be included to plasticity rules of a similar spirit as the one we used here. Spatial cell types like grid and head direction cells are widely believed to follow specific connectivity implementing a continuous attractor [69–71]. Indeed, typical continuous attractor models are characterized by their fixed connectivity patterns, establishing an attracting manifold as well as a velocity modulated update mechanism thereon. Projections from CA1 to EC would then putatively implement a predictive mechanism that would be focussed on predicting the outcome of this rigid update mechanism using the predictive representation of the conjunctive spatial and non spatial features obtained from the hippocampal loop. However, as this is a slightly different kind of model, there might not be a straightforward result linking the weights to a successor representation type quantity, but this would be an interesting avenue for further computational work.

Finally, we want to mention that the hippocampal formation is also of high interest in studying generalization in human cognitive neuroscience [72,73]. Since evidence is growing that the systems which are partaking in spatial representations are also recruited to encode more abstract variables, possibly forming 'conceptual spaces' [74–76], it might be interesting to understand whether an inherent bias for symmetry also shapes these representations. In particular, one might speculate for which domains of cognition such a bias for symmetry or a metric space structure might be adequate, and when it would not be. This might for example be tested by constructing explicit tasks where dynamics are reversible and compare subjects accuracy in predictions to such tasks where the dynamics are not.

In conclusion, our model contributes to the theoretical framework of hippocampal predictive representations, both on the mechanistic level through suggesting the use of symmetric local learning rules, as well as on the functional level, where such learning might be useful for generalization in spatial learning.

## 4. Methods and materials

### 4.1. Methods

#### 4.1.1. Neural network model.
We consider two populations of rate-based neurons $p^1 \in \mathbb{R}^m$, $p^2 \in \mathbb{R}^n$ respectively. The population $p^1$ is recurrently connected via a matrix of synaptic weights $W^r \in \mathbb{R}^{m \times m}$, and feeds its activity forward to population $p^2$ via the matrix of synaptic weights $W^f \in \mathbb{R}^{n \times m}$. Both regions receive additional external inputs, $\phi^1, \phi^2$, and decay to equilibrium in the absence input. The temporal evolution of the populations is then

modelled by the following differential equations:

$$\frac{d}{dt}p^1 = -p^1 + \sigma\left(\gamma_1 W^r p^1 + (1 - \gamma_1)\phi^1\right) \tag{27}$$

$$\frac{d}{dt}p^2 = -p^2 + \sigma\left(\gamma_2 W^f p^1 + (1 - \gamma_2)\phi^2\right). \tag{28}$$

Here, $\sigma$ is an activation function, and the $\gamma$ parameters control the relative weight of the different types of inputs. On the computational level, these parameters will correspond to the discounting factor of the successor representation, effectively scaling how far in the future predictions are made.

On the level of biological implementation, there are two speculative interpretations one may have for these parameters. First, the $\gamma_i$ might be associated with acetylcholine. Indeed, this transmitter is believed to control the relative strength of external and internal input in hippocampus [77]. In particular, a low value of $\gamma$ would correspond to an encoding mode, where the network is relatively stronger driven by external input and thus susceptible to encoding associations or transition structures between these inputs. For high $\gamma$, the network is in retrieval mode and its outputs will depend more on the recurrent weights and therefore previously learned transition structure. Indeed, this interpretation has been pursued in [25], who showed how a recurrent net could learn weights at low $\gamma$ and then use those learned weights to retrieve successor representations at arbitrary high $\gamma$. Another interpretation of $\gamma$ would be that it represents the respective proportions of inputs of different kinds a certain subpopulation receives due to anatomic difference. That is, it could for example encode the number of synaptic contacts from other CA3 axons for a certain subpopulation - then understood not in absolute terms, but compared to other cells of the same type. Indeed, there is evidence that both connections in as well as connections to hippocampus are not homogeneous but vary along the septotemporal and proximodistal axis [78,79]. Thus, it might be plausible to assume a varying degree of strength of the different input types. In this work, however, we do not explicitly model these biological details, and thus the precise role of $\gamma$ remains undetermined.

Assuming the weights (and the external inputs) change on a slower timescale than the population dynamics, we can let the above differential equations go to equilibrium for analysis - in practice, we can also integrate the ODE above with a timescale $\tau$ orders of magnitude smaller than the timescale of learning. If $\sigma$ is approximately linear, this results in

$$p^1 = (1 - \gamma_1)(\mathrm{Id} - \gamma_1 W^r)^{-1}\phi^1 \tag{29}$$

$$p^2 = \gamma_2 W^f p^1 + (1 - \gamma_2)\phi^2. \tag{30}$$

We assume that the animal receives inputs depending on the current state of the environment, which we will denote by the process $S_t$. In the reinforcement learning setting, the agent influences the way in which the state-process is sampled by selecting actions $A_t$, but at this point this is not relevant since we are only building up a predictive representation of states, not actions. The inputs to the two neural populations thus take the form

$$\phi_t^i = \phi^j(S_t) \tag{31}$$

where $\phi^i$ is a function mapping the state space to an activation pattern in neural space. If the state space is discrete, we can also write this as

$$\phi_t^i = \Phi^i e_{S_t} \tag{32}$$

with $\Phi^i$ now being a matrix of appropriate dimension, and $e_{S_t}$ is the unit-vector corresponding to state $S_t$, which hence selects the corresponding activation pattern from $\Phi^i$s columns. In the special case $\Phi^i = \text{Id}$, we have a so called one-hot encoding, where hence every cell corresponds to a distinct state and fires if and only if the agent is in that state.

**4.1.2. Learning rule.** The synaptic weights - both feedforward and recurrent - in our model are subject to a learning rule which is controlled by two parameters $\alpha, \beta \in \mathbb{R}$:

$$\Delta W = \alpha \left( p_{t+1}^{post} - W p_t^{pre} \right) p_t^{pre\,T} + \beta \left( p_t^{post} - W p_{t+1}^{pre} \right) p_{t+1}^{pre\,T}. \tag{33}$$

Here, $p^{pre}, p^{post}$ denote the vectors of pre- and postsynaptic activities. The parameters $\alpha, \beta$ do not model a biological substrate, they control the qualitative behaviour of the learning rule: they serve as a (crude) discrete approximation for a continuous plasticity kernel as is typically observed in spike-time dependent plasticity protocols. Indeed, for $\alpha = 1$ the learning rule strengthens connections where post-synaptic activity at the later timestep was high (measured relative to overall input), and presynaptic activity at the earlier timestep was high as well. In contrast, for $\beta = 1$, the learning rule strengthens connections where postsynaptic activity at an earlier timestep was high, and so was presynaptic activity at the later timestep.

Note that a synaptic update of the form

$$\Delta W = (p_{post} - W p_{pre}) p_{pre}^T. \tag{34}$$

may be seen as performing a sort of conditional expectation or regression objective: At equilibrium, the expected update in synaptic weights, given the current activity, should be zero

$$\mathbb{E}\left[ \Delta W | p_{pre} \right] = 0. \tag{35}$$

We have that this holds in general only if

$$W p_{pre} = \mathbb{E}\left[ p_{post} | p_{pre} \right]. \tag{36}$$

That is, if $W$ encodes an optimal prediction of the post-synaptic activity given the presynaptic activity. Hence, heuristically, $\alpha$ emphasizes to learn weights that are optimally tuned to predict the next state of their output. In our setting, the cells are driven by external input identifying the state $S_t$ of the world, so this amounts to optimally predicting the next state $S_{t+1}$. In turn, $\beta$ emphasizes a backward prediction, where $S_t$ should be predicted by $S_{t+1}$.

## 4.2. Forward and backward process, reversibility

Recall that a time-homogeneous Markov process on a finite state space $\mathcal{S}$ is determined by its one-step transition probabilities

$$P_{s,s'} := p(S_{t+1} = s' | S_t = s) = p(S_1 = s' | S_0 = s). \tag{37}$$

Under standard assumptions, the process has a unique stationary distribution [80,81], that is, a probability distribution $\pi$ over states such that

$$\sum_s \pi(s)p(S_1 = s'|S_0 = s) = \pi(s'). \tag{38}$$

That is, if we are in stationary distribution, the probability mass is left invariant when transitioning according to our process. In particular, if we assume we start in stationary distribution, then $p(S_t = s) = \pi(s)\forall t$. We may now define the reverse process of our original process, by simply reversing the order of time. That is, define a process $R_t$ with transition probabilities

$$p(R_{t+1} = s'|R_t = s) = p(S_t = s'|S_{t+1} = s). \tag{39}$$

These probabilities will generally depend on the timestep $t$, but they will not if our process $S_t$ is in stationary distribution. In that case, we again have a single matrix that encodes all transition probabilities:

$$P_{s,s'}^{backward} = p(R_1 = s'|R_0 = s) \tag{40}$$
$$= p(S_0 = s'|S_1 = s) \tag{41}$$
$$= p(S_1 = s|S_0 = s')\frac{\pi(s)}{\pi(s')}. \tag{42}$$

In matrix notation, we have

$$P^{backward} = \Pi^{-1}P^T\Pi, \tag{43}$$

where $\Pi$ is a diagonal matrix with the values of $\pi$ on the diagonal. It is easy to see from this definition that $P^{backward}$ is again a valid transition matrix, and that it has the same stationary distribution as $P = P^{forward}$.

Forward and backward transition probabilities are equal if and only if the process fullfills the *detailed balance* condition

$$\pi(s)P_{s,s'} = \pi(s')P_{s',s}. \tag{44}$$

In that case, the process is called reversible. In (Eq 19) we have used a weighted sum of transition probabilities of the forward and backward process Note that the symmetrized process with transition probabilities

$$P^{sym} = \frac{1}{2}(P^{forward} + P^{backward}) \tag{45}$$

will always be reversible, as can either be seen by a simple algebraic calculation, or by the following argument: Let $P^*$ be the *adjoint* of $P$ with respect to the inner product induced by $\pi$. This means that for any vectors $x,y$ we have

$$x^T\Pi Py = (P^*(x))^T\Pi y. \tag{46}$$

We then have that

$$P^* = P^{backward}, \tag{47}$$

as may be seen by plugging in the definition of $P^{backward}$ into (Eq 46). For the adjoint, we have that $(P^*)^* = P$. Thus, in particular,

$$(P^{sym})^* = \frac{1}{2}(P + P^*)^* = \frac{1}{2}(P^* + P) = P^{sym}. \tag{48}$$

This also explains that it is warranted to talk about a 'symmetrization' here: indeed, the adjoint with respect to the standard inner product is the transpose of a matrix $A$, in which case one would obtain the standard symmetric part of a matrix, that is, $\frac{1}{2}(A + A^T)$.

## 4.3. Reinforcement learning and successor features

In Reinforcement learning, one generally considers a Markov decision process (MDP), that is a tuple $(\mathcal{S}, \mathcal{A}, T, R, \gamma)$, where $\mathcal{S}, \mathcal{A}$ are the sets of possible states and actions, $T(s'|s, a)$ gives the transition probability from state $s$ to state $s'$ when choosing action $a$, $R(s,a)$ is obtained reward when chosing action $a$ in state $s$, and $\gamma$ is a discount factor. The goal in RL usually is to find an optimal policy, that is a probability distribution $\pi(a|s)$ over actions, given states, which maximizes the expected discounted cumulative reward

$$\mathbb{E}\left[\sum_{k=0}^{\infty} \gamma^k R(S_t, A_t)\right], \tag{49}$$

where the states and actions $S_t, A_t$ are sampled according to the policy $\pi$ and the transition probabilities $T$. In our experiments, we are only interested in the particularly simple case where the transitions are deterministic - that is, taking an action $a$ in state $s$ surely leads to a specified state $s'(a)$. Furthermore, we only consider the situation where the reward is a function of the state only. However, all definitions in the paragraph below readily generalize to functions of states and actions. Assume a fixed policy $\pi$ and a process $S_t$ following said policy.

Now consider any mapping from the state space

$$\phi : \mathcal{S} \to \mathbb{R}^m, \tag{50}$$

which we can interpret as an observable generated by the state space. Then define a function on the state space

$$SF_\phi^t(s) = \mathbb{E}\left[\sum_{k=0}^{\infty} \gamma^k \phi(S_{t+k}) | S_t = s\right]. \tag{51}$$

$SF_\phi$ thus gives the expected (exponentially weighted) cumulative future sum of the observation or feature $\phi$, given the current state $s$. This is hence called a 'successor feature' in the literature. Define $G_t = \sum_{k=0}^{\infty} \gamma^k \phi(S_{t+k})$. We then have

$$SF_\phi^t(s) = \mathbb{E}[G_t | S_t = s] = \mathbb{E}\left[\phi(S_t) + \sum_{k=1}^{\infty} \gamma^k \phi(S_{t+k}) | S_t = s\right] \tag{52}$$

$$= \mathbb{E}[\phi(S_t) + \gamma G_{t+1} | S_t = s] \tag{53}$$

$$= \mathbb{E}[\phi(S_t) | S_t = s] + \gamma \mathbb{E}[\mathbb{E}[G_{t+1} | S_{t+1}] | S_t = s] \tag{54}$$

$$= \phi(s) + \sum_{a,s'} \mathbb{P}[S_{t+1} = s' | S_t = s] \gamma SF_\phi^{t+1}(s'). \tag{55}$$

In particular, if the transition probabilities of $S_t$ are time-homogeneous, we see that $SF$ itself does not depend on $t$, and we can write

$$SF_\phi(s) = \mathbb{E}\left[\phi(S_t) + SF_\phi(S_{t+1})|S_t = s\right]. \tag{56}$$

Temporal Difference learning (TD-learning) uses this relationship to construct a target to update an estimate of $SF_\phi$ online. Indeed, if $\tilde{SF}_t$ is the current estimate the agent has for the successor features, then consider update rules

$$\tilde{SF}_{t+1}(S_t) = \tilde{SF}_t(S_t) + \varepsilon\Delta_t \tag{57}$$

$$\Delta_t = \phi(S_t) + \gamma\tilde{SF}_t(S_{t+1}) - \tilde{SF}_t(S_t). \tag{58}$$

Then we see that if $\tilde{SF}_t = SF_\phi$, we get $\mathbb{E}[\Delta_t|S_t = s] = 0$. Thus, here $\phi(S_t) + \gamma\tilde{SF}_t(S_{t+1})$ is used as a bootstrapping estimate of the target $\mathbb{E}[\phi(S_t) + \gamma\tilde{SF}_t(S_{t+1})|S_t = s]$, and $\tilde{SF}_t$ is updated according to the error to that target - in total, $\Delta_t$ is thus also called TD-error.

Successor features encompass important special case examples: For the choice of $\phi = R$ the reward function, $SF_R$ becomes the value function: The value function under a policy $\pi$, which is typically denoted as $V^\pi$, hence encodes the weighted cumulative sum of expected rewards, given the current state. In particular, the current estimate of the reward function may be used to define a new policy as

$$\pi^*(\cdot|s) = \delta_{a^*}$$
$$a^* = \arg\max_a V^\pi(s'(a)). \tag{59}$$

That is, the policy deterministically selects the action $a^*$ which leads to the next state with the highest value according to the current value function estimate. Iterating this process then successively learns a better value function - this process of updating an estimate of the value function and then choosing an optimal policy with respect to it is the basis of the classical TD-learning algorithm [38]. Indeed, for our navigation experiments we use this approach, only replacing the maximum in (Eq 59) by a softmax which smoothens the transition probabilities. Note that the above assumes a model of which actions lead to which next states - which posing as given should be a sensible assumption in navigation problems - but a completely model-free approach simply computes the value of a state-action pair instead.

**Successor Representations.** In the case that we have that $\phi$ is in fact an injective mapping, we can define a modified version of successor features as

$$SR_\phi(\rho) = \mathbb{E}\left[\sum_{k=0}^\infty \gamma^k\phi(S_{t+k})|\phi(S_t) = \rho\right], \tag{60}$$

where $\rho \in \phi(\mathcal{S})$. Here, the injectivity of the mapping is necessary to ensure $SR$ defined as above still enjoys desirable properties like homogenity in time and the Bellman equation, but besides of that one could also define a similar quantity without using injectivity. Now, in the injective situation, we would like to call the above 'successor representation'. This is because when we take mapping $\phi(s) = e_s$, where $e_S$ is a unit vector in $\mathbb{R}^{|S|}$, then we obtain

$$SR_\phi(e_s) = (Id - \gamma P)_s^{-1} \tag{61}$$

which corresponds to the classical definition of successor representation. In general, if we assume $\Phi \in \mathbb{R}^{m \times |S|}$ is the matrix of features (could also be an operator if we allow for continuous state space), then we have that

$$SR_\phi(\phi(s)) = (\Phi(Id - \gamma P)^{-1})_s = (\Phi(Id - \gamma P)^{-1}\Phi^-)\phi(s),  \quad (62)$$

where $\Phi^-$ is such that $\Phi^-\Phi = Id_{|S|}$.

**Generalization and Successor Features.** The idea behind using successor features for some set of function $\phi^i$, instead of simply directly computing the the value function is that of generalization/transfer: Assume the reward function $R$ can be written as a linear combination of the features $\phi$, that is

$$R = \sum_i w_i \phi^i.  \quad (63)$$

Then also for the respective predictive representations one has

$$SF_R(s) = \sum_i w_i SF_\phi(s).  \quad (64)$$

Now assuming that the reward changes to a new function $\tilde{R}$, which also can be expressed with the features, then the only thing that has to be relearned is the weights $w_i$, while the successor features $SF_\phi$ can be reused. Thus, one may then generalize more easily to a new task, since the transition structure under a policy is essentially already learnt. In particular, in a discrete state space, a fixed reward function can $R$ can of course be encoded by a reward vector $\mathbf{R}$, that is

$$\mathbf{R} = \sum_s R(s)e_s.  \quad (65)$$

The value function is then simply given through the classical SR as

$$V(s) = (Id - \gamma P)^{-1} \mathbf{R}(s).  \quad (66)$$

Thus, for a fixed policy, this separates the computation of value into learning a successor representation and learning a reward vector. Our navigation experiments probe the generalization capability of this approach, by introducing a new reward vector $\tilde{\mathbf{R}}$, after an optimal policy and a SR for a reward vector $\mathbf{R}$ have been learned. Importantly, only the reward vector is allowed to be relearned, while the SR has to be reused from the previous task.

**Symmetric TD-learning.** In our reinforcement learning experiments, we use a modified version of TD-learning to mimic the behaviour of the local learning rule in the SR-network. In practice, Successor features are typically parametrized by some parameter $\theta$ (e.g., the weights of a neural network), which is then updated to reduce the TD-error. That is, for each value of $\theta$ we obtain a map $\tilde{SF}_\theta(s)$ of states. We can then update the parameter $\theta$ via

$$\theta_{t+1} = \theta_t + \varepsilon\Delta(\theta)_t  \quad (67)$$

$$\Delta(\theta)_t = \alpha\left(\phi(S_t) + \gamma\tilde{SF}_{\theta_t}(S_{t+1}) - \tilde{SF}_{\theta_t}(S_t)\right)^T \nabla_\theta \tilde{SF}_{\theta_t}(S_t)  \quad (68)$$

$$+ \beta\left(\phi(S_{t+1}) + \gamma\tilde{SF}_{\theta_t}(S_t) - \tilde{SF}_{\theta_t}(S_{t+1})\right)^T \nabla_\theta \tilde{SF}_{\theta_t}(S_{t+1}),  \quad (69)$$

where $\alpha, \beta$ are parameters corresponding to the ones controlling the local learning rule. In particular, for $\alpha = 1, \beta = 0$ one obtains the classical TD-learning rule for function approximation [38] and for $\alpha = 0, \beta = 1$ one obtains the 'predecessor representation' [16]. The case $\alpha = \beta$ yields a symmetrized version of TD-learning. In our experiments, we use a particularly simple version of the above: for a discrete state space, one can simply parametrize $\tilde{SF}$ by means of a matrix $M \in \mathbb{R}^{d \times k}$, where $d$ is the number of features and $k$ is the number of states. Then one has $\tilde{SF}_M(s) = Me_s$, and hence the update rule

$$\Delta(M)_t = \alpha \left( \phi(S_t)e_{S_t}^T + \gamma Me_{S_{t+1}}e_{S_t}^T - Me_{S_t}e_{S_t}^T \right)^T \tag{70}$$
$$+ \beta \left( \phi(S_{t+1})e_{S_{t+1}}^T + \gamma Me_{S_t}e_{S_{t+1}}^T - Me_{S_{t+1}}e_{S_{t+1}}^T \right).$$

## 4.4. Successor representations and hitting times

In the theory of Markov chains, it is a common exercise to study the fundamental matrix of the Markov process, which encodes properties about the process via the first hitting times of states [82]. The first hitting time of a state, $\tau_1^s$, is defined as the first time a process hits the state $s$. The successor representation may similarly be interpreted as an operator that encodes certain expectations related to the first hitting times. Indeed, in Appendix H in S1 Appendices we prove the following formula:

$$M_{ss'} = \delta(s, s') + \begin{cases} 0, \mathcal{T}(s'|s) = 0 \\ \mathbb{E}[\gamma^{\tau_1^{s'}}|S_0 = s], \mathcal{T}(s'|s) = 1 \\ \frac{\mathbb{E}[\gamma^{\tau_1^{s'}}]|S_0=s]}{1-\mathbb{E}[\gamma^{\tau_1^{s'}}]|S_0=s']}, \mathcal{T}(s'|s) = \infty. \end{cases} \tag{71}$$

Here, $\mathcal{T}(s'|s)$ denotes the largest number of times which state $s'$ may be hit with nonzero probability when starting from state $s$. It may only take values 0, if $s'$ may not be hit from $s$ at all, 1, if $s'$ is a transient state, or $\infty$, when the process may arbitrarily often return to a state. (Eq 71) provides a convenient reformulation of the value function. In particular, when we are in the setting of a navigation task, where the goal is navigating to target $s^*$, the reward is a unit reward at $s^*$. Hence, for an optimal policy $\pi^*$ which only chooses among shortest paths to the target, the successor representation becomes

$$M_{ss'} = \delta(s, s') + \begin{cases} 0, s = s' \neq s^* \\ 0, s \neq s', s' \text{not on a shortest path from } s \text{ to } s^* \text{ which the policy visits} \\ \gamma^{d(s,s')}, s' \neq s^* \neq s \\ \frac{\gamma^{d(s,s')}}{1-\gamma^{d(s,s')}}, s' = s^* \end{cases} \tag{72}$$

where $d(s, s')$ is the distance on the state space. The value function may then be read of as the column corresponding to $s^*$. We use this formula in Appendix H in S1 Appendices to prove a stability result of optimal policies in the navigational setting under symmetrization: If we are in the deterministic navigation setting, and our policy $\pi$ is optimal, then after taking the symmetrization of the transition probabilities and computing the value function with these probabilities, the policy $\pi$ is still optimal.

### 4.5. A predictive representation leads to backward shift only in the asymmetric case

Here we want to give an analytic explanation for the stronger backward shifts of the centre of mass observed in Fig 4 when using an asymmetric learning rule, while a symmetric learning rule does not result in a shift. Recall that we simulated an agent repeatedly running on a linear track, always in the same direction, which was then reset to the starting position. The effect is most easily analyzed if we ignore boundaries and go to a continuous situation. Let's thus assume an agent is repeatedly running on the real line $\mathbb{R}$. In the beginning, before learning, the cells in our model are just driven by the external input, and thus can be taken equal to the features. That is, cell $j$ will take the value $\phi^j(x)$ when at position $x$. After learning, it will instead encode a successor feature. With continuous space it is also more convenient to assume continuous time. The successor representation may be easily transferred to this setting [23], where the definition then is

$$SF_{\phi^j}(x) := \int_0^\infty e^{-\gamma t} \int \phi^j(y) p_t(y|x) \, dy \, dt, \tag{73}$$

where $p_t(y|x)$ is the probability of transitioning from state $x$ to state $y$ in time $t$. The centre of mass (COM) of a cells' firing field is just the spatial mean of the cells' activity when the latter is normalized to be a probability distribution. That is, for example before learning, when the cell $j$ fires at position $x$ according to the feature $\phi^j$, the $j$-th centre of mass is

$$COM(\phi^j) = \frac{\int_{\mathbb{R}} x \phi^j(x) \, dx}{\int_{\mathbb{R}} \phi^j(x) \, dx}. \tag{74}$$

To understand the observed shifting effect in $COM$, it is now instructive to study the successor feature with deterministic dynamics: Assume we are moving with constant velocity $v$, that is,

$$p_t(y|x) = \delta(y - (x + tv)), \tag{75}$$

then the successor feature takes the form

$$SF_{\phi^j}(x) = \int_0^\infty e^{-\gamma t} \phi^j(x + tv) \, dt. \tag{76}$$

Now, taking the mean over all positions yields:

$$\int x SF_{\phi^j}(x) \, dx = \int x \int_0^\infty e^{-\gamma t} \phi^j(x + tv) \, dt \, dx \tag{77}$$

$$= \int_0^\infty x \int e^{-\gamma t} \phi^j(x + tv) \, dx \, dt \tag{78}$$

$$= \int_0^\infty (u - tv) \int e^{-\gamma t} \phi^j(u) \, du \, dt \tag{79}$$

$$= \int_0^\infty u \int e^{-\gamma t} \phi^j(u) \, du \, dt - \int_0^\infty tv \int e^{-\gamma t} \phi^j(u) \, du \, dt \tag{80}$$

$$= \frac{1}{\gamma} \int u \phi^j(u) \, du - \frac{v}{\gamma^2} \int \phi^j(u) \, du. \tag{81}$$

Similarly we have

$$\int SF_{\phi^j}(x)dx = \int \int_0^\infty e^{-\gamma t}\phi^j(x+tv)dtdx \tag{82}$$

$$= \int \int_0^\infty e^{-\gamma t}\phi^j(u)dtdu \tag{83}$$

$$= \frac{1}{\gamma}\int \phi^j(x)dx. \tag{84}$$

Thus, in total we obtain

$$COM(SF_{\phi^j}) = \frac{\int u\phi^j(u)du}{\int \phi^j(u)du} - \frac{v}{\gamma} = COM(\phi^j) - \frac{v}{\gamma} \tag{85}$$

That is, the center of mass is shifted backward by a factor which is controlled by the prediction timescale $\gamma$ and the velocity of the movement. We thereby also obtain the seemingly new prediction that faster running speed and smaller terminal place field size (as a proxy for $\gamma$) should result in bigger shifts. Now if our forward process is a deterministic process with constant velocity $v$, then conceptually, our backward process is also deterministic with velocity $-v$ (although stricly speaking, there is no stationary distribution). In particular, the symmetrized process is a process which in an infinitesimal amount of time travels either forward with velocity $v$ or backward with velocity $v$ with equal probability. The shifts thus cancel and the centre of mass stays the same as for the initial feature. The same relation of shift and velocity/timescale holds also when the dynamics come from a Brownian motion with constant drift, that is $dx_t = vx_t + \sqrt{\rho}B_t$, as we show in Appendix B in S1 Appendices. Now when we symmetrize the Brownian motion with drift, the drift term will disappear, that is, we have a standard Brownian motion. It is thus clear that under this motion, there should again be no shift of the centre of mass.

## 4.6. Experiments

## 4.7. Convergence experiments

For our initial convergence experiments (Fig 3, top row) we simulate our model in a circular random walk setting with 30 states and use $n = 40$ cells in each layer. The input at each state is drawn from an i.i.d. Gaussian distribution ($\sigma = 0.1$) (i.e., for each cell and each state we draw a value from a Gaussian distribution, this is then assumed to be the input for that cell whenever the specific state is visited). Throughout this and the following experiments, we use a moderately high value of $\gamma = 0.7$ - meaning a relatively large timescale/small discounting. This is an arbitrary choice, but relatively high values have been used in the past in SR-theories of hippocampus [83].

We observe 100000 transitions, and repeat the experiment 30 times. We then repeat the same experiment (top row, right of Fig 3), also drawing the transition matrix $P$ randomly. To track the convergences, we consider the loss terms

$$\mathcal{L}W_r = ||\mathbb{E}[\Delta W_r]|| \tag{86}$$

$$\mathcal{L}W_f = ||\mathbb{E}[\Delta W_f]||, \tag{87}$$

where the terms on the right hand vanish at convergence and are defined in (Eq 45). The resulting convergence curves are shown in Fig 3. These experiments are conducted with a linear activation function, we also repeat this experiment with the activation functions **tanh**, **relu** as shown in S1 Fig.

To check the effect of different values $\alpha, \beta$, we conduct a parameter sweep in a circular random walk, and random Gaussian features as above. For each combination $\alpha, \beta$ we run 30 random initializations for 1000 iterations, and check this combination once for the recurrent layer and once for the feedforward layer. Here, we track convergence similar as above, but now we take the fraction

$$\frac{\mathcal{L}W}{\mathcal{L}W_0} \tag{88}$$

with $W_0$ the initial weight, to judge if we are moving towards or away from equilibrium. This results in the matrices shown in Fig 3. To speed up the experiments, we use a smaller network for these experiments, with 20 neurons in each layer and 10 states. Again we use $dt = 0.1$.

## 4.8. Linear track experiments

To simulate the animal running on a linear track, we partitioned a track of length $300cm$ in 50 discrete states and let the agent perform a rightward biased walk that either took a step to the right ($p = 0.9$) or stayed in place ($p = 0.1$). We chose a time step of $dt = 0.4$, corresponding to a velocity of $\approx 15cm/s$. The agent would run 25 laps on the linear track, with a short resting phase in between two laps. We set up a two-layer model, with $n = 100$ cells in each layer, and again $\gamma = 0.7$. For the feedforward weights, we always used the parameters $\alpha = 1, \beta = 0$, while for the recurrent weights we tested both the symmetric case $\alpha = \beta = \frac{1}{2}$ as well as the asymmetric case $\alpha = 1, \beta = 0$. We then performed an analysis approach as in [37]. For each lap, we collected the mean activities of each layer per state and computed the center of mass (COM) by the formula

$$\text{COM}_j = \frac{\sum_x x \bar{p}_j(x)}{\sum_x \bar{p}_j(x)} \tag{89}$$

where $x$ is position on the track and $\bar{p}_j(x)$ denotes the mean activity of cell $j$ when at that location. To obtain a distribution of the overall shifts of the COMs over all laps, we calculated the average COMs for the first and the last five laps and subtracted them from each other, obtaining the histograms in Fig 4. To track the evolution of the shift, we subtracted the COMs from the 12-th lap, which yields a more gradual tracking of shifts. We plot the results of this approach in the scatterplots in Fig 4.

## 4.9. Reinforcement learning

We investigated the generalization capabilities of a symmetric over an asymmetric learning rule in TD-learning in different scenarios.

**4.9.1. Navigation experiments.** Since the hippocampal formation is prominently involved in navigation tasks, we first focused on tasks of such nature. We thus studied different variations of the same general problem setup: Given a grid environment with a deterministic transition structure, the agent would receive a unit reward only when arriving at the designated target state $s_T$, upon which the episode is terminated, and the next episode starts in an initial state $s_0$, drawn uniformly at random. The agent is thus encouraged to navigate from all possible starting states to the target state to receive a reward. We used the Neuronav toolbox [84] to implement the grid environment and implemented our modified SR-agent in the

framework of that toolbox. For completeness, let us here again give a concise summary of how the agent learns. The agent possesses a world-model $P(s'|s, a)$ and updates an internal estimate $M$ of the successor representation as well as an estimate $w$ of the reward vector. Having observed a transition from state $s$ to state $s'$ via action $a$ and a reward $r$, the updates are

$$\Delta(M) = \alpha \left( e_s e_s^T + \gamma M e_{s'} e_s^T - M e_s e_s^T \right) \tag{90}$$

$$+ \beta \left( e_{s'} e_{s'}^T + \gamma M e_s e_{s'}^T - M e_{s'} e_{s'}^T \right) \tag{91}$$

$$\Delta(w) = (r - w(s)) e_s. \tag{92}$$

In summary, this is how the agent selects actions and updates its internal estimates:

- Given the current state $s$ and current estimate $M$ for the successor representation, compute $q(s, a) = \sum_{s', s''} w(s'') M(s''|s') P(s'|s, a)$
- Select an action according to $\pi(a|s) = \text{softmax}(q(s, a))$
- Observe next state $s'$ and reward $r$
- Update reward estimate $w$ and successor representation estimate $M$.

To check for generalization capability, we would train agents which utilize the respective learning rules (symmetric,asymmetric) to navigate to a fixed $s_T$ first. We trained the agent either for a fixed number of episodes (we used 400 episodes with a maximum number of steps of 400), or until a fixed accuracy criterion was met (the mean deviation from optimal performance for the preceding 8 episodes was lesser than 2). Then, we would randomly select a new target state $\tilde{s}_T$ from among all possible states. Importantly, after the modification of the target states, the agents were only allowed to modify the reward-prediction vector $w$, but not the successor representations $M$ themselves. That is, they could only learn the reward structure of the new task, but had to rely on the transition structure that was encoded during learning the previous task, which also depends on the policy. We repeated this experiment for 200 times, with randomly drawn combinations $(s_T, \tilde{s}_T)$ in each repetition. The results of these experiments are depicted in Fig 5. For the first experiments reported in the main text, we used a learning rate of 0.1 and set $\gamma = 0.7$. We chose a relatively high value of $\gamma$, as this is typically done when modelling hippocampal place cells in standard navigation tasks [8,20], but we repeated the second kind of analysis for different values of $\gamma$ and the learning rate, as shown in S6 Fig.

For the experiments in the maze, we used the same parameters for the maximum number of steps and the performance criterion, but higher values for the learning rate and the discount factor: $\gamma = 0.9, lr = 0.999$. This is because for lower parameter values the agents needed too many repetitions to converge.

Since all the environments we tested in the above setting have a symmetric nature, that is, their underlying state space is an undirected graph, we repeated the same generalization experiment in a setting where the state space is a directed graph, and hence travel time between two nodes is not symmetric. We constructed a simple graph with 17 nodes, which is essentially a tree graph with the addition of a directed edge from the lowest level to the highest. On this graph, we performed the same kind of navigation experiment. We used the same values for the learning rate and $\gamma$ (0.1,0.7), but reduced the number of episodes to 50 since the state space is smaller and hence can be learned faster.

**4.9.2. Policy entropy.** In Fig 6 we computed the entropy of the current policy of the agents after each episode of learning. As explained previously, we computed the current policy

by applying the softmax-function to the q-values

$$q(s, a) = \sum_{s', s''} w(s'') M(s''|s') p(s'|a, s) \tag{93}$$

where $w$ is the estimate of the reward vector and $M$ is the current estimate of the successor representation. Then the policy is: $\pi(a|s) = \text{softmax}(q(s, a))$, and we computed the entropy of the policy averaged over all states as

$$H[\pi] = \frac{1}{|\mathcal{S}|} \sum_{s \in \mathcal{S}} \sum_a -\ln \pi(a|s) \pi(a|s). \tag{94}$$

## Supporting information

**S1 Appendices. Contains mathematical proofs and additional explanation for statements made in the main text.**
(PDF)

**S1 Fig. Convergence experiments with different activation functions.** We conducted the same experiments as in Fig 3, with the activation functions (tanh, relu) from left to right.
(EPS)

**S2 Fig. Grid world environments used in navigation tasks.** Outline of the environments used to produce Fig 5. In all environments, agents started at random locations and could choose between four actions (or stay in place when a move would hit a wall). Plots were generated using the Neuronav package [84].
(EPS)

**S3 Fig. Generalization performance in individual environments with fixed number of episodes.** Generalization performance in the individual environments that are averaged to generate the left plot in Fig 5.
(EPS)

**S4 Fig. Generalization performance in individual environments with fixed accuracy criterion.** Generalization performance in the individual environments that are averaged to generate the right plot in Fig 5.
(EPS)

**S5 Fig. Generalization when seeing the same data while training.** This figure is identical to Fig 4.9.2, except that here we trained the symmetric agent on the transitions sampled by the asymmetric agent—hence both agents see exactly the same data before generalization.
(EPS)

**S6 Fig. Generalization performance for varying choices of discount factors and learning rates.** All experiments were conducted in the 'empty' environment. Note that in the case where the asymmetric agent outperforms the symmetric agent, generalization is considerably worse than in the regimes where it does not.
(EPS)

## Acknowledgments

We thank William de Cothi, Tom George and Nikola Milosevic for helpful discussions.

## Author contributions

**Conceptualization:** Janis Keck, Caswell Barry, Christian F. Doeller, Jürgen Jost.

**Data curation:** Janis Keck.

**Formal analysis:** Janis Keck.

**Funding acquisition:** Caswell Barry, Christian F. Doeller, Jürgen Jost.

**Investigation:** Janis Keck.

**Methodology:** Janis Keck.

**Project administration:** Janis Keck.

**Resources:** Caswell Barry, Christian F. Doeller, Jürgen Jost.

**Software:** Janis Keck.

**Supervision:** Caswell Barry, Christian F. Doeller, Jürgen Jost.

**Validation:** Janis Keck.

**Visualization:** Janis Keck.

**Writing – original draft:** Janis Keck.

**Writing – review & editing:** Janis Keck, Caswell Barry, Christian F. Doeller, Jürgen Jost.

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
