## [Decision Letter · Decision Letter 0]

15 Dec 2024

PCOMPBIOL-D-24-01249

Impact of symmetry in local learning rules on predictive neural representations and generalization in spatial navigation

PLOS Computational Biology

Dear Dr. Keck,

Thank you for submitting your manuscript to PLOS Computational Biology. After careful consideration, we feel that it has merit but does not fully meet PLOS Computational Biology's publication criteria as it currently stands. Therefore, we invite you to submit a revised version of the manuscript that addresses the points raised during the review process.

Please submit your revised manuscript within 60 days Feb 14 2025 11:59PM. If you will need more time than this to complete your revisions, please reply to this message or contact the journal office at ploscompbiol@plos.org. Please include the following items when submitting your revised manuscript:

We look forward to receiving your revised manuscript.

Kind regards,

Suhita Nadkarni, Ph.D

Academic Editor

PLOS Computational Biology

Daniele Marinazzo

Section Editor

PLOS Computational Biology

**Journal Requirements:**

At this stage, the following Authors/Authors require contributions: Janis Samuel Keck, Caswell Barry, Christian F. Doeller, and Jürgen Jost. Please ensure that the full contributions of each author are acknowledged in the "Add/Edit/Remove Authors" section of our submission form.

2) We ask that a manuscript source file is provided at Revision. Please upload your manuscript file as a .doc, .docx, .rtf or .tex. If you are providing a .tex file, please upload it under the item type LaTeX Source File and leave your .pdf version as the item type Manuscript.

4) We notice that your supplementary Figures are included in the manuscript file. Please remove them and upload them with the file type 'Supporting Information'. Please ensure that each Supporting Information file has a legend listed in the manuscript after the references list.

5) Some material included in your submission may be copyrighted. According to PLOS copyright policy, authors who use figures or other material (e.g., graphics, clipart, maps) from another author or copyright holder must demonstrate or obtain permission to publish this material under the Creative Commons Attribution 4.0 International (CC BY 4.0) License used by PLOS journals. Please closely review the details of PLOS copyright requirements here: PLOS Licenses and Copyright. If you need to request permissions from a copyright holder, you may use PLOS's Copyright Content Permission form.

Potential Copyright Issues:

i) Figure 4 :Thank you for stating that “figure adapted from Dong, C., Madar, A. D., & Sheffield, M. E. (2021).” Please give appropriate credit to the original author(s) and the source, provide a link to the Creative Commons license, and indicate if changes were made.

7) Your current Financial Disclosure states, " JK, JJ and CFD are supported by the Max Planck Society and the Max Planck School of Cognition.CB is funded by a Wellcome SRF."

However, your funding information on the submission form indicates five funders. Please ensure that the funders and grant numbers match between the Financial Disclosure field and the Funding Information tab in your submission form. Note that the funders must be provided in the same order in both places as well.                                                  . 

Please indicate by return email the full and correct funding information for your study and confirm the order in which funding contributions should appear. Please be sure to indicate whether the funders played any role in the study design, data collection and analysis, decision to publish, or preparation of the manuscript.

**Reviewers' comments:**

Reviewer's Responses to Questions

Reviewer #1: I provide my comments to improve the presentation of the manuscript.

Strengths:

Insight into Hippocampal Function: The study provides a novel perspective on how local learning rules in CA3 and CA1 might contribute to differential representations for spatial navigation, which is a significant contribution to hippocampal research.

Theoretical Modeling and Simulation: The authors' analytical framework, supported by simulation, effectively shows how symmetric and asymmetric rules shape the successor representation (SR) and generalization abilities in navigation tasks, connecting well to observed hippocampal dynamics.

Symmetry as a Functional Mechanism: The study’s findings on how symmetry aids generalization in symmetric environments add a functional perspective to the theoretical SR model, making it highly relevant for applications in reinforcement learning.

Weaknesses and Suggestions for Improvement:

Biological Plausibility of the Model: While the model simulates CA3 and CA1 structures, it oversimplifies the biological processes in the hippocampus, notably lacking feedback from the entorhinal cortex (EC) and plasticity at multiple synaptic levels. Integrating a loop structure to include EC interactions and plasticity at feedforward and recurrent connections would enhance the model's relevance to hippocampal function. However, I appreciate that incorporating these changes may not be trivial. Therefore, perhaps the authors could address this aspect in the discussion.

Limitations of Binary Symmetry Conditions: The paper’s binary classification of learning rules as strictly symmetric or asymmetric could be too simplistic. Real-world neural networks may have a spectrum of rule dynamics. The authors could explore a gradient or mixed rule approach, investigating how variations between pure symmetry and asymmetry impact SR and generalization performance.

Empirical Validation: Although the model replicates certain observed CA3 and CA1 characteristics, the study is primarily theoretical. It would benefit from empirical tests, such as in vivo or in vitro validation of place cell responses under manipulated learning conditions. This would strengthen the biological relevance of their findings.

Environmental Complexity: The navigation tasks used are relatively simple, which limits the generalizability of the conclusions to real-world or complex spatial environments. Adding a variety of more complex or dynamically changing environments (e.g., variable paths or obstacle courses) would better test the robustness and practical applicability of the SR under symmetric versus asymmetric rules.

Parameter Interpretability and Sensitivity Analysis: Parameters like the discount factor (γ) and the symmetry coefficients (α and β) are crucial for the SR, but their biological analogs are speculative. Further clarification or sensitivity analysis on these parameters, particularly γ in relation to neural phenomena (e.g., neuromodulatory influences like acetylcholine), would improve interpretability and model alignment with biological data.

The study advances understanding of the role of local learning rule symmetry in predictive representations and spatial navigation. However, addressing the biological realism, expanding to more complex navigational scenarios, and providing empirical support would significantly improve the paper's rigor and applicability to both neuroscience and AI.

Reviewer #2: # Impact of symmetry in local learning rules on predictive neural representations and generalization in spatial navigation

In this work, the authors extend the existing work on successor representation models of the hippocampus and more precisely to the CA3-CA3 recurrent synapses. In short, this works contributes to hippocampal predictive representations based on symmetric local learning rules.

This works provides a solid mathematical study of the dynamics of a two populations model of CA3 and CA1 neurons where the CA3 population feeds the CA1 one with plastic connections described by the Markov process $(p,W)$. The inputs to the CA3 population are assumed to be realizations of a time homogenous Markov process. Through the use of stochastic analysis, the authors are able to study the stationary distribution of $(p,W)$ and apply these results to (for example) classical place field experiments.

Although I really appreciate this work, I do not think it is suitable for publication in the current for for the reasons I elaborate below. I think having a mathematical study of a "simple" paradigm in biology is very good way to do science because there are some simple cases that are proved to be true. In the realm of biology, these are very rare and valuable landmarks. I also appreciate the incredible amount of work of the paper: modeling, mathematics and numerics, more that 50 pages...

## Major revision

There are two ways to read the paper. Either you discard the proofs (and thus this appendices) or you read everything.

- If you discard the mathematical developments of the appendices, it is nearly impossible to follow the analysis. The equations are spread out in the entire documents (appendices) and not all notations, formulas, etc can be found.

- If you choose to read the appendix and follow the mathematics thoroughly, then they are several gaps which prevent from having a complete understanding of the developments. This is mainly because, the equations are not properly spelt out and it is very difficult to follow some of the arguments (see below in minors revisions).

It think it is better to be overly verbose (and repeat equations).

- section 2.3. At this stage, most readers are lost. What is the dynamics for $p_t$ (I guess 24-25), which weight $W^f$, $W^r$ is affected by plasticity?

Obviously, I have read everything. Appendix A is not really understandable. Appendix B is OK. Appendix D is kind of OK despite D.1 being not. The proof of th.1 is good. Appendix E is OK. Appendix F is good: it is the amount of details, precision that I would expect.

## minor revision

- in the abstract, change "center of mass" into "mean"?

- author summary "symmetric in time" -> "symmetric wrt to exchange of input timings"?

- page 3, typo "temporal difference(TD)"

- page 3, you mention Bi and Poo for the CA3-CA1 synapse but Bi and Poo focused on cultured neurons

- page 5, are you sure about (4)?

- page 5 what are $\lambda$ in (5-6)?

- page 6: Figure `C. I find very surprising the asymmetric learning rule case. You have the same input but DeltaW is completely different

- page 21 I do not understand eq 36-37. Where is $\tilde SF_t(S)$ defined?

- page 22 above (40). Are you sure $e_s$ is injective?

- page 33 type "retrieves A.More"

- page 33, notation (64) is not very clear.

- Equation 66 is very difficult to find. Please make a lemma and prove it.

- theorem 1 page 37, what is alpha?

- appendix D.1 What is $\Pi$? Please report the equations of the model fully.

- prop 2 page 26. typo (Id - gamma P)^-1) . This typo can be found elsewhere

- type eq (185)

**Have the authors made all data and (if applicable) computational code underlying the findings in their manuscript fully available?**

Reviewer #1: Yes

Reviewer #2: Yes

PLOS authors have the option to publish the peer review history of their article (what does this mean?). If published, this will include your full peer review and any attached files.

Reviewer #1: No

Reviewer #2: No

**Figure resubmission:**
---

## [Decision Letter · Decision Letter 1]

14 Apr 2025

Dear Dr. Janis Samuel Keck

We are pleased to inform you that your manuscript 'Impact of symmetry in local learning rules on predictive neural representations and generalization in spatial navigation' has been provisionally accepted for publication in PLOS Computational Biology.

Best regards,

Suhita Nadkarni, Ph.D

Academic Editor

PLOS Computational Biology

Daniele Marinazzo

Section Editor

PLOS Computational Biology

Reviewer's Responses to Questions

**Comments to the Authors:**

Reviewer #1: The authors have addressed all my comments in the revised manuscript.

Reviewer #2: The authors have addressed most of my points in this intensively revised manuscript. They also corrected mistakes I did not catch.

This paper is much longer now but I sincerely think it will help the curious reader. As I said, I enjoyed the paper and the paradigm which is considered.

Hence, I think the paper is suitable for publication in the current form.

**Have the authors made all data and (if applicable) computational code underlying the findings in their manuscript fully available?**

Reviewer #1: Yes

Reviewer #2: Yes

PLOS authors have the option to publish the peer review history of their article (what does this mean?). If published, this will include your full peer review and any attached files.

Reviewer #1: No

Reviewer #2: No

---

## [Editor Report · Acceptance letter]

PCOMPBIOL-D-24-01249R1

Impact of symmetry in local learning rules on predictive neural representations and generalization in spatial navigation

Dear Dr Keck,

I am pleased to inform you that your manuscript has been formally accepted for publication in PLOS Computational Biology. Your manuscript is now with our production department and you will be notified of the publication date in due course.

With kind regards,

Anita Estes
